
**Evaluation of NO+ reagent ion chemistry for on-line measurements of atmospheric volatile**
**organic compounds**
Abigail R. Koss[1,2,3], Carsten Warneke[1,2], Bin Yuan[1,2], Matthew M. Coggon[1,2], Patrick R. Veres[1,2],
Joost A. de Gouw[1,2,3]
*1. NOAA Earth System Research Laboratory (ESRL), Chemical Sciences Division, Boulder, CO,*
*USA*
*2. Cooperative Institute for Research in Environmental Sciences, University of Colorado at*
*Boulder, Boulder, CO, USA*
*3. Department of Chemistry and Biochemistry, University of Colorado at Boulder, CO, USA*



**Abstract**
NO$^+$ chemical ionization mass spectrometry (NO$^+$ CIMS) can achieve fast (sub 1-Hz) on-
line measurement of trace atmospheric volatile organic compounds (VOCs) that cannot be ionized
with H$_3$O$^+$ ions (e.g. in a PTR-MS or H$_3$O$^+$ CIMS instrument). Here we describe the adaptation of
a high-resolution time-of-flight H$_3$O$^+$ CIMS instrument to use NO$^+$ primary ion chemistry. We
evaluate the NO$^+$ technique with respect to compound specificity, sensitivity, and VOC species
measured compared to H$_3$O$^+$. The evaluation is established by a series of experiments including
laboratory investigation using a gas-chromatography (GC) interface, in-situ measurement of urban
air using a GC interface, and direct in-situ measurement of urban air. The main findings are that
(1) NO$^+$ is useful for isomerically resolved measurements of carbonyl species; (2) NO$^+$ can achieve
sensitive detection of small (C4-C8) branched alkanes, but is not unambiguous for most; and (3)
compound-specific measurement of some alkanes, especially iso-pentane, methylpentanes, and
high mass (C12-C15) n-alkanes, is possible with NO$^+$. We also demonstrate fast in-situ chemically
specific measurements of C12 to C15 alkanes in ambient air.
Keywords: NO$^+$, chemical ionization mass spectrometry, VOCs, atmosphere, PTRMS





## 1. Introduction

Volatile organic compounds (VOCs) are central to the formation of ozone and secondary organic aerosol (SOA), and can have direct human health effects. Understanding the behavior of these species in the troposphere presents several measurement challenges (Glasius and Goldstein, 2016). First, VOCs are highly chemically diverse. Second, many environmentally important species require measurement precision of better than 100 parts-per-trillion (ppt). Finally, numerous applications, such as eddy flux analyses or sampling from a mobile platform, require fast in-situ measurements, with sub-1 minute time resolution.

$H_3O^+$ chemical ionization mass spectrometry ($H_3O^+$ CIMS) , more commonly known as proton-transfer-reaction mass spectrometry (PTR-MS), is a well-established approach to measuring VOCs (de Gouw and Warneke, 2007;Jordan et al., 2009b). In $H_3O^+$ CIMS, air is mixed with hydronium ($H_3O^+$) ions in a drift tube region. VOCs are ionized by transfer of the proton from $H_3O^+$ to the VOC. These instruments are capable of VOC measurements that are fast, sensitive, and chemically detailed (Jordan et al., 2009b;Graus et al., 2010;Sulzer et al., 2014;Yuan et al., 2016).

Despite these advantages, $H_3O^+$ CIMS has several limitations related to the reagent ion chemistry. For one, this technique generally cannot distinguish between isomers. For instance, this is a significant limitation when measuring aldehyde and ketone carbonyl isomers, which have very different behavior in the atmosphere. Separation of propanal and acetone with PTRMS has been explored using collision-induced dissociation with an ion-trap mass analyzer, but this technique negatively affects the instrument time resolution and sensitivity (Warneke et al., 2005). Additionally, some proton transfer reactions are dissociative. Large hydrocarbons (C8 and larger) fragment into common small masses, making spectra difficult to interpret (Jobson et al., 2005;Erickson et al., 2014;Gueneron et al., 2015). Alcohols and aldehydes can lose $H_2O$, lowering the sensitivity to the protonated parent mass; their product ion masses then coincide with those of hydrocarbons, making independent measurement difficult (Španěl et al., 1997;Buhr et al., 2002). Furthermore, $H_3O^+$ CIMS is not sensitive to small (~C8 and smaller) saturated alkanes, as their proton affinities are lower than or very close to that of water (Arnold et al., 1998;Gueneron et al., 2015). This is a serious limitation in studies of urban air or emissions from oil and natural gas extractions, where small alkanes can contribute a large fraction to the total gas phase carbon and



chemical reactivity (Katzenstein et al., 2003;Gilman et al., 2013). Gas chromatography techniques
avoid many of these limitations, but have much slower time resolution.

Use of $NO^+$ reagent ion chemistry may address some of the limitations of $H_3O^+$. Reaction

of $NO^+$ with various VOCs has been extensively studied using selected-ion flow tube methods
(SIFT-MS). SIFT methods use a quadrupole mass filter in between the ion source and ion-molecule
reactor, which provides a very pure reagent ion source but limits the primary ion signal. SIFT
studies have identified the major products of the reaction of $NO^+$ with VOCs representative of
many different functional groups (Španěl and Smith, 1996, 1998a, b, 1999;Španěl et al.,
1997;Arnold et al., 1998;Francis et al., 2007a;Francis et al., 2007b). Aldehydes and ketones are
easily separable: ketones cluster with $NO^+$, forming mass ($m$+30) ions, whereas aldehydes react
by hydride abstraction, forming mass ($m$-1) ions (where $m$ is the molecular mass of the species).
Rather than losing $H_2O$, as in $H_3O^+$ CIMS, alcohols react by $NO^+$ adduct formation or hydride
abstraction. And finally, $NO^+$ can be used to detect alkanes: small (>C4) branched alkanes and
large (>C8) n-alkanes react by hydride abstraction, forming mass ($m$-1).

The application of SIFT methods to atmospheric analysis has been limited by relatively

poor sensitivity (Smith and Španěl, 2005;Francis et al., 2007b;de Gouw and Warneke, 2007);
although better sensitivities have been reported in recent years (Prince et al., 2010). The adaptation
of an existing CIMS instrument to use the SIFT technique requires extensive instrument
modification or the purchase of an external SIFT unit (Karl et al., 2012). Several groups have
experimented with low-cost adaptation of $H_3O^+$ CIMS instruments to use $NO^+$ chemistry.
Knighton et al. (2009) adapted an $H_3O^+$ CIMS instrument to measure 1,3-butadiene and
demonstrated in-situ detection of this species in the atmosphere. Jordan et al. (2009a) have
developed a hollow-cathode ion source capable of switchable reagent ion chemistry, and
demonstrated laboratory measurement with $NO^+$ of several aromatics, chlorinated aromatics, and
carbonyls, with sensitivities comparable to $H_3O^+$ CIMS. The $NO^+$ capability of the Jordan et al.
instrument has been used in the laboratory by Inomata et al. (2013) to investigate detection of n-
tridecane and by Agarwal et al. (2014) to measure picric acid, and by Liu et al. (2013) to investigate
the behavior of MVK and methacrolein in a reaction chamber.

These studies suggest that an easy, low-cost adaptation of $H_3O^+$ CIMS instruments to $NO^+$

chemistry could greatly enhance our capability to measure VOCs in the atmosphere. However, the
number of VOC species investigated to-date is small and few field measurements have been



reported. The ability of a modified $H_3O^+$ CIMS instrument to separate carbonyl isomers in ambient
air, and to measure small alkanes both in the laboratory and in ambient air, has not been evaluated.
Finally, the lack of fragmentation of n-tridecane reported in Inomata et al. (2013) is intriguing, but
the use of an $NO^+$ CIMS instrument to measure similar high-mass alkanes in ambient air has not
been demonstrated.
Here we evaluate the adaptation of an $H_3O^+$ CIMS instrument to use $NO^+$ reagent ion
chemistry. We provide specifics on instrument set-up and operating parameters. We report the
sensitivity and spectral simplicity of $NO^+$ CIMS, relative to $H_3O^+$ CIMS, for nearly 100
atmospherically relevant VOCs, including a wide range of functional groups, and provide product
ion distributions for several representative compounds. We demonstrate, interpret, and evaluate
measurements of separate aldehyde and ketone isomers, light alkanes, and several other species in
ambient air. Finally, we investigate measurement of high-molecular-mass alkanes using $NO^+$. We
extend the laboratory analysis of high-mass alkanes to C12-C15 n-alkanes and demonstrate fast,
in-situ measurement of these species in ambient air.
**2. Methods**
**2.1 Instrumentation**
Two separate $H_3O^+$ CIMS instruments (referred to hereafter as PTR-QMS and $H_3O^+$ ToF-
CIMS) were adapted to NO+ chemistry in this work. Both instruments consist of (1) a hollow
cathode reagent ion source, (2) a drift tube reaction region, (3) an ion transfer stage that transports
from the drift tube to the mass analyzer and allows differential pumping, and (4) a mass analyzer.
Both instruments have nearly identical hollow cathode ion sources and drift tube reaction regions,
described in detail in de Gouw and Warneke (2007). The PTR-QMS (Ionicon Analytik) uses ion
lenses to transfer ions from the drift tube to a unit-mass-resolution quadrupole mass analyzer
(Pfeiffer). This instrument is described further by de Gouw and Warneke (2007). The $H_3O^+$ ToF-
CIMS uses RF-only segmented quadrupole ion guides to transfer ions from the drift tube to a time-
of-flight mass analyzer with a mass resolution of 4000-6000 produced by Aerodyne Research Inc.
/ Tofwerk (Bertram et al., 2011). This instrument is described further by Yuan et al. (2016). A
similar PTR-ToF instrument using quadrupole ion guides has also been recently described (Sulzer
et al., 2014).





A gas chromatograph (GC) instrument was used both as an interface to the ToF-CIMS and
as a separate instrument using an electron-impact quadrupole mass spectrometer. The GC collects
VOCs in a liquid nitrogen cryotrap for a 5 minute period every 30 minutes. VOCs are then injected
onto parallel $Al_2O_3$/KCl PLOT and semi-polar DB-624 capillary columns to separate C2-C11
hydrocarbons and heteroatom-containing VOCs. When used as an interface to the ToF-CIMS, the
column eluant was directed to the inlet of the ToF-CIMS, where it was diluted with 50 sccm of
dry clean air. When operated as a separate instrument, the column eluant was directed to an
electron-ionization quadrupole mass spectrometer (EIMS) operated in selected-ion mode. The
response of this GC-EIMS instrument to various VOCs has been well characterized over a long
period of field and laboratory applications, and further operational details have been reported
elsewhere (Goldan et al., 2004;Gilman et al., 2010;Gilman et al., 2013).

**2.2 Adaptation of $H_3O^+$ to $NO^+$ CIMS.**
Ideally, both $H_3O^+$ and $NO^+$ reagent ion chemistry can be utilized with a single instrument.
The fewest possible number of hardware parameters were changed to facilitate fast switching and
instrument stability.
To achieve generation of $NO^+$ ions, the water reservoir was replaced with ultra-high purity
air. The source gas flow (5 sccm), the hollow cathode parameters, and the drift tube operating
pressure (2.4mbar) were not changed. To optimize the generation of $NO^+$ ions relative to $H_3O^+$,
$O_2^+$, and $NO_2^+$, and the generation of the desired $VOC^+$ ion products, the voltages of the
intermediate chamber plates, $V_{IC1}$ and $V_{IC2}$, and the drift tube voltage $V_{DT}$ were adjusted. An
instrument schematic showing the locations of $V_{IC1}$, $V_{IC2}$, and $V_{DT}$ can be found in the
supplementary information (Fig. S1). Optimization was performed sampling dry air.
It has been demonstrated that the quadrupole ion guides of the ToF-CIMS can significantly
change the measured distribution of reagent and impurity ions (Yuan et al., 2016). The PTR-QMS
does not have that issue as strongly and therefore we explored the effect of $V_{IC1}$, $V_{IC2}$, and $V_{DT}$ on
reagent ion distribution using the PTR-QMS. As the PTR-QMS and ToF-CIMS have nearly
identical ion source and drift tube design, we assume that ion behavior in these regions is the same
for the two instruments.
First, $V_{DT}$ was held constant at 720V (the original setting of the PTR-QMS instrument),
and $V_{IC1}$ and $V_{IC2}$ were varied (Fig. 1). The settings of $V_{IC1}$ (140V) and $V_{IC2}$ (80V) were selected





as a compromise between high $NO^+$ ion count rate and low impurity ion count rates. The major
impurity ions are $H_3O^+$, $O_2^+$, and $NO_2^+$, and it is desirable to limit the formation of these ions
because they react with VOCs, complicating the interpretation of spectra. Next, several VOCs with
different functional groups were introduced into the instrument, separately, and the drift tube
electric potential scanned. A drift tube voltage of 350 V (electric field intensity relative to gas
number density E/N= 60 Td) was selected as a compromise between maximizing $NO^+$ ion count
rate, minimizing $H_3O^+$, $O_2^+$, and $NO_2^+$, maximizing VOC ion count rates, minimizing alkane
fragmentation, and promoting different product ions for carbonyls and aldehydes (Fig. 2). This
setting results in about 10e6 cps of $NO^+$ primary ions, while in typical PTR-MS settings we achieve
about 30e6 cps of $H_3O^+$ primary ions.
We note that the E/N of 60 Td used for the $NO^+$ CIMS is much lower than that used in
typical PTRMS settings (circa 120 Td). In air, $NO^+$ will react with water to produce $H_3O^+$ and
$HNO_2$ (Fehsenfeld et al., 1971). The electric field in the drift tube limits the formation of the
$NO^+\cdot(H_2O)_n$ intermediaries in this reaction, promoting high $NO^+$ count rates and VOC sensitivity.
In PTRMS, the drift field is used to prevent the formation of analogous $H_3O^+\cdot(H_2O)_n$ clusters. The
bond energy of $H_3O^+\cdot(H_2O)_n$ clusters is significantly higher than that of $NO^+\cdot(H_2O)_n$ clusters
(Keesee and Castleman, 1986), hence the need for a higher E/N in PTRMS settings.
The remainder of the work detailed in this manuscript was performed using the ToF-CIMS
with the settings as described here. The ToF-CIMS has the advantages of high mass resolution,
fast time resolution, and simultaneous measurement of all masses. Further small adjustments were
made to the ToF-CIMS quadrupole ion guide voltages using Thuner software (Tofwerk AG) to
promote sensitivity to VOCs and separate carbonyl isomers.
**3. Results and Discussion**
**3.1 Laboratory experiments**
**3.1.1 Sensitivity and simplicity of the $NO^+$ reagent ion chemistry**
VOCs from several calibration cylinders (VOCs listed in Table S1) were diluted with high
purity air to mixing ratios of approximately 10 ppbv, and introduced into the sampling inlet of the
GC interface. Eluant from the column was directed into the ToF CIMS as described above. Several
species co-elute with another compound (m- and p- xylenes; myrcene and camphene; 1-ethyl,3-



methylbenzene and 1-ethyl,4-methylbenzene); reported sensitivities and product ions are an average of the two co-eluting species.

Each VOC mixture was sampled twice, once with $H_3O^+$ and once with $NO^+$ reagent ion chemistry and instrument settings. Based on the results we evaluated the utility of $NO^+$ CIMS relative to $H_3O^+$ CIMS using two metrics. The first metric is sensitivity for individual VOCs. To determine the sensitivity ($S$), the signals (counts per second) of all product ions were integrated over the width of the chromatographic peak and sensitivities for the measured VOCs using $NO^+$ chemistry were calculated relative to the sensitivity using $H_3O^+$ chemistry ($S_{NO+}/S_{H3O+}$). For several VOCs, we also calculated the relative sensitivity if only the most abundant product ion (the quantitation ion) is measured (Table 2B).

The second metric is the simplicity of spectra. In an ideal instrument, each VOC would produce only one product ion, and each ion mass would be produced by only one VOC. However, using $NO^+$ and $H_3O^+$ reagent ions, fragmentation of product ions does occur. As a metric for the complexity of the product ion distribution resulting from particular VOCs, we determined the fraction of the most abundant ion to the total signal from this VOC ($F$) and discuss ($F_{NO+}$) relative to ($F_{H3O+}$). Figure S2 contains a comparison of $F_{NO+}$ and $F_{H3O+}$, and an example product ion distribution. A larger value of this ratio means that $NO^+$ reagent ion chemistry creates a simpler product ion distribution for that particular VOC. This metric does not indicate whether a particular product ion is produced by only one VOC. Uniqueness of product ions is discussed in Sect. 3.1.2. The $NO^+$ CIMS product ion distributions of 25 atmospherically relevant VOCs are reported in Table 2.

Figure 3 summarizes the comparison between $NO^+$ and $H_3O^+$ reagent ion chemistry for the two metrics. On the y-axis the spectrum simplicity metric and on the x-axis the sensitivity metric are shown.

Branched alkanes and most cyclic alkanes are detected with far greater sensitivity using $NO^+$ chemical ionization than with $H_3O^+$ chemical ionization. Aromatics and alkenes are detected slightly more sensitively, and, on average, ketones are detected slightly less sensitively. Alcohols are detected more sensitively, by at least a factor of two, with the exception of methanol. The lower sensitivity to methanol is consistent with slower reaction kinetics reported in the literature (Španěl and Smith, 1997). Monoterpenes and acetonitrile are detected substantially less sensitively.



In comparing the simplicity of the product ion distribution between $H_3O^+$ and $NO^+$
chemistry, most branched and cyclic alkanes, ketones, and monoterpenes have a higher fraction of
signal on a single product ion (simpler spectra). We also highlight that many alkyl substituted
aromatics fragment substantially with $H_3O^+$ chemistry but do not with $NO^+$ chemistry. The few
exceptions (notably, benzene) create more complicated spectra because an $NO^+$ cluster product is
also present (*m+30*).

### 212 3.1.2 Distribution of product ions

Product ions of C4-C10 alkenes, aldehydes, ketones, alcohols, and aromatics are consistent
with product ion distributions and mechanisms reported from SIFT investigations. Reaction
mechanism (charge transfer, hydride transfer, or cluster formation) is dependent on the
thermodynamics of charge transfer and hydride transfer (Fig. 4, Table 1, values from Lias et al.
(1988)). Charge transfer occurs if the reaction enthalpy is favorable, regardless of the hydride
transfer enthalpy. If the charge transfer enthalpy is close to zero, then $NO^+$ clustering occurs; and
if charge transfer is not favorable but hydride transfer is, then hydride transfer will occur.
Carbonyls participate in two mechanisms: ketones cluster with $NO^+$, and aldehydes hydride
transfer. Branched alkanes exclusively undergo hydride transfer, and other functional groups
participate in other mechanisms: aromatics undergo charge transfer and occasionally cluster;
alcohols undergo hydride transfer, and alkenes charge transfer, cluster, or hydride transfer
depending on the size of the molecule and the location of the double bond within the molecule.

### 225 3.1.3 Alkane fragmentation

Small (C4-C10) branched alkanes cannot be measured by $H_3O^+$ CIMS. With $NO^+$ CIMS,
these VOCs are detectable but generally fragment to produce several ionic fragments that are
common to different species. These masses (for example, m/z 57 $C_4H_9^+$) are produced by many
different compounds and are likely not useful for chemically resolved atmospheric measurements.
A few masses (e.g. m/z 71 $C_5H_{11}^+$ and m/z 85 $C_6H_{13}^+$) are only produced by a few compounds and
were therefore targeted for further investigation in ambient air measurements. Conversely, cyclic
alkanes fragment very little. Fig. 5 shows the product ion distributions of several representative
aliphatic compounds. We note that the major product ions of cyclic alkanes *(M-H)* are the same
with $H_3O^+$ and with $NO^+$ chemistry. However, the mechanism is different: $NO^+$ ionizes by hydride
abstraction, while $H_3O^+$ ionizes by protonation followed by loss of $H_2$ (Midey et al., 2003). The
$H_3O^+$ ionization mechanism has a secondary channel consisting of protonation followed by





elimination of $CH_4$ or $C_nH_{2n}$ (Midey et al., 2003). The difference in ionization mechanism is a
likely explanation for the lower degree of fragmentation observed using $NO^+$ chemistry.

Compared to small (C8 and smaller) alkanes, large (C12 and higher) n-alkanes show little

fragmentation, with at least 50% of the total ion signal accounted for by the expected parent mass
($m$-$1$) (Fig. 6). Additionally, the degree of fragmentation decreases with increasing carbon chain
length. It is quite difficult to measure these compounds with $H_3O^+$ CIMS because they fragment
extensively and are not detected sensitively (Erickson et al., 2014). $NO^+$ CIMS could provide a
fast, sensitive, chemically specific measurement of these compounds. It should be mentioned that
large n-alkanes (C10 and larger) are not measureable with the GC interface. Dodecane ($C_{12}H_{26}$),
tridecane ($C_{13}H_{28}$), tetradecane ($C_{14}H_{30}$), and pentadecane ($C_{15}H_{32}$) were sampled directly with the
$NO^+$ ToF-CIMS and product ions were identified by correlation with the expected major product
ion ($m$-$1$). The $NO^+$ ToF-CIMS sensitivity to pentadecane was determined using a permeation
source (Veres et al., 2010).

**3.1.4 Instrument response factor for select compounds**

A calibration factor was determined for various VOCs by (1) direct calibration, (2) estimation from
sensitivity relative to $H_3O^+$ CIMS, or (3) estimation from correlation with GC-EIMS (Table 2).
Direct calibrations were performed by mixing a known concentration of a VOC from either a
permeation cell (pentadecane) or a calibration gas cylinder (other VOCs) into a dry high-purity air
dilution stream. Calibration factors estimated from sensitivity relative to $H_3O^+$ CIMS were
calculated using $H_3O^+$ ToF-CIMS calibration factors and results from laboratory GC-CIMS
experiments (Sect. 3.1.1). Calibration factors for $H_3O^+$ ToF-CIMS were determined in previous
work (Yuan et al., 2016). These calibration factors were multiplied by the relative peak areas
determined in Sect. 3.1.1 to obtain estimated $NO^+$ ToF-CIMS calibration factors. (An example
chromatogram and calculation is shown in Fig. S3). Calibration factors estimated from correlation
with GC-EIMS were calculated from the slope of $NO^+$ Tof-CIMS measurements against GC-EIMS
measurements in ambient air (discussed in further detail in Sect. 3.2.2).

In the following discussion we use two metrics of instrument response: counts-per-second

(cps) and normalized counts-per-second (ncps). Counts-per-second (cps) is the raw ion count rate
of the instrument. Two operations were applied to cps measurements to obtain ncps. First, a duty
cycle correction (*d.c.c.*) was applied (Chernushevich et al., 2001):



$$I_{corr} = cps \times \sqrt{\frac{m/z_{reference}}{m/z}} \qquad (1)$$

where $I_{corr}$ is the duty-cycle corrected ion count rate and $m/z_{reference}$ is an arbitrary reference mass
(in this work $m/z_{reference} \equiv 55$). The duty-cycle correction accounts for differences in ion residence
time in the extraction region of the ToF and eliminates a mass-dependent sensitivity bias. Then,
measurements were normalized to the duty-cycle corrected $NO^+$ (primary ion) measurement,
which typically has count rates on the order of $10^6$ above that of VOCs:
$$ncps = 10^6 \frac{I_{corr}}{NO^+_{corr}} \qquad (2)$$

The normalization removes variability due to fluctuations in the ion source and detector. In
calculating limits of detection, we use duty-cycle uncorrected cps, as this best reflects the
fundamental counting statistics of the instrument. In reporting ambient air measurements, we use
ncps. The ncps measurement reduces several significant instrumental biases and better reflects
VOC abundances in air.

Limits of detection at 1Hz measurement frequency were calculated by finding the mixing

ratio at which the signal-to-noise ratio (S/N) is equal to 3. The calculation can be expressed by
(Bertram et al., 2011; Yuan et al., 2016):
$$\frac{S}{N} = 3 = \frac{C_f[X]_{lod}t}{\alpha \times \sqrt{C_f[X]_{lod}t + 2Bt}} \qquad (3)$$

where $C_f$ is the instrument response factor, in cps per ppb; $[X]_{lod}$ is the limit-of-detection

mixing ratio of species X in ppb; $t$ is the sampling period of 1 second; $\alpha$ is the scaling factor of
noise compared to expected Poissonian counting statistics; and $B$ is the background count rate in
cps. The scaling factor $\alpha$ is generally greater than 1 because high-resolution peak overlap and
fitting algorithms create additional noise (Cubison and Jimenez, 2015). For comparison, $H_3O^+$
ToF-CIMS limits of detection, using the same ToF-CIMS instrument, are included where
available.

Aliphatics and aromatics are generally detected quite sensitively. Aromatics have sub-100

ppt detection limits and are detected slightly more sensitively with $NO^+$ CIMS than with $H_3O^+$
CIMS, with $NO^+$ detection limits generally about 30% lower. Aliphatic species are detected with
quite low detection limits (less than 50 ppt) and with substantially better sensitivity than $H_3O^+$: the
detection limit of methylcyclohexane using $NO^+$ is a factor of 27 lower than with $H_3O^+$.





Aldehydes and ketones also have detection limits of around 100 ppt or less, with the
exception of acetaldehyde (l.o.d. = 355 ppt). The higher detection limit of acetaldehyde is due to
a somewhat higher instrumental background and a lower response factor that is consistent with
reaction kinetics (Španěl et al., 1997). Methanol has a very high detection limit (28 ppb); this is
expected from the anomalously low rate constant of the methanol-$NO^+$ reaction (Španěl and Smith,
1997). In contrast, ethanol is detected far more sensitively with $NO^+$ than with $H_3O^+$, with a
detection limit of 105 ppt (compared to 1600 ppt for $H_3O^+$).
**3.1.5 Humidity dependence**
Humidity-dependent behaviors of primary ions and selected VOCs (acetaldehyde, acetone,
isoprene, 2-butanone, benzene, toluene, o-xylene, and 1,3,5-trimethylbenzene) were determined
by diluting a VOC calibration standard into humidified air to reach approximately 10ppb mixing
ratio, then sampling directly with the $NO^+$ ToF-CIMS. Air temperature was 27°C. Product ion and
signal dependences on humidity for selected primary ions and VOCs are shown in Fig. 7
(additional species are included in Fig. S4). As relative humidity increases, $NO^+$ (m/z 30) remains
relatively constant, while protonated water and protonated water clusters (especially m/z 37,
$H_5O_2^+$) increase. As the abundance of $H_3O^+$ in the drift tube increases, one might expect to see
increased products of VOC reaction with $H_3O^+$ with a corresponding decrease in $NO^+$ products.
Although an increase of $H_3O^+$ product is seen for some species (e.g. MEK), it is not universally
true. For many species, the major effect is that the $NO^+$ adduct product increases relative to other
$NO^+$ product ions. This effect is especially intense for isoprene, where the isoprene-$NO^+$ cluster
(m/z 98, $C_5H_8NO^+$) increases by a factor of 10 from 0 to 70% relative humidity. A similar humidity
effect, observed during SIFT measurements of alkenes, has been reported previously by Diskin et
al. (2002), who attributed the effect to better stabilization of excited intermediary $(NO^+ \cdot R)^*$ ions
by $H_2O$. A full investigation of this effect is beyond the scope of this manuscript. In lieu of a
complete theoretical understanding of humidity effects, we suggest that an experimental humidity
correction could be applied as in Yuan et al. (2016).
**3.2 Measurements of urban air**
**3.2.1 GC-$NO^+$ CIMS measurements**
Measurement of ambient air using the GC interface allowed us to determine which
compounds in ambient air produce which masses. This is the essential link between laboratory
measurements of calibration standards, and interpretation of ambient $NO^+$ ToF-CIMS
measurements. Ambient air from outside the laboratory was sampled from Oct. 27, 2015-Oct. 30,
2015 through an inlet three meters above ground level, and directed through 10 meters of ½"
diameter Teflon tubing at a flow rate of 17 slpm (residence time approximately 4 seconds). The
GC interface subsampled this stream. Eluant from the column was directed into the $NO^+$ ToF
CIMS as described in Sect. 2.1. The laboratory is in an urban area (Boulder, CO) and the inlet was
located near a parking lot and loading dock. Instrument background (including the GC interface)
was determined by sampling zero air at the beginning and end of each measurement period.
Instrument performance and stability, and retention times of selected compounds, were checked at
least once per day by sampling a 56-component hydrocarbon calibration standard.

Figure 8 shows several masses from a typical chromatogram. In this chromatogram, it is

clear, for instance, that the majority of signal from m/z 83 ($C_6H_{11}^+$) can be attributed to one
compound (methylcyclopentane). On the other hand, m/z 57 ($C_4H_9^+$) is produced from many
different compounds with comparable intensities. Aldehydes and ketones appear to be well
separated, as expected from the laboratory experiments. Figure 9 summarizes the contributions of
different VOCs to several ions (m/z 57, $C_4H_9^+$ and m/z 83 $C_6H_{11}^+$) during the entire measurement
period. M/z 57 ($C_4H_9^+$) has contributions from many different VOCs, and the relative proportions
are highly variable. Conversely, m/z 83 ($C_6H_{11}^+$) is mostly attributable to methylcyclopentane
during the majority of the measurement period. M/z 57 ($C_4H_9^+$) does not provide a useful
measurement of alkanes, while m/z 83 ($C_6H_{11}^+$) may possibly provide a useful measurement of
methylcyclopentane. Corresponding figures for other masses can be found in the supplemental
information (Fig. S5-S7). Table 3 summarizes our assessment of key ions.
**3.2.2 $NO^+$ CIMS vs. GC-EIMS Measurement Comparison**

Measurements using the GC interface do not provide any information about the fast time

response capability of the $NO^+$ ToF-CIMS. Additionally, not all compounds detectable by $NO^+$
CIMS and present in ambient air can be transmitted through the GC interface. Simultaneous GC-
EIMS and $NO^+$ ToF-CIMS measurements were conducted to investigate fast $NO^+$ measurements,
determine if there are any significant interferences to key $NO^+$ masses, and explore $NO^+$ CIMS
response to VOCs not transmittable through the GC interface. Ion masses that are produced by
VOCs not detectable with the GC have higher and more variable signal when measured by the
$NO^+$ ToF-CIMS, compared to the GC-ToF-CIMS.



Ambient air was sampled into the laboratory as described in the previous section. The GC-
EIMS and the NO$^+$ ToF-CIMS were run as separate instruments and subsampled the 17 SLPM
flow at the same point. Measurements were taken from Nov. 4, 2015 through Nov. 6, 2015. The
GC-EIMS instrument was operated on a 30-minute schedule. Instrument background was
determined from zeros taken at the beginning and end of the measurement period. The 56-
component hydrocarbon calibration standard was sampled once per day. The NO$^+$ ToF-CIMS
measured at 1 Hz frequency. Instrument zeros were taken for a two minute period once every hour.
Calibration gas from a 10-component hydrocarbon standard was sampled for two minutes once
every three hours. At the end of the measurement period, both instruments were disconnected from
the ambient air line and sampled air from inside the laboratory for 1.5 hours (three GC samples),
to investigate the NO$^+$ ToF-CIMS response to air with a VOC composition substantially different
from urban air.
For all comparisons between the two instruments, the 1Hz NO$^+$ ToF-CIMS measurements
were averaged over the 5-minute GC-EIMS collection period. The NO$^+$ ToF-CIMS was re-
calibrated using air with ambient humidity for the 10 species listed in Table 2A, and no further
humidity correction was applied. Correlations between independent GC and calibrated CIMS
measurements generally show high correlation coefficient ($R^2 > 0.9$) and slopes close to 1
(examples in Fig. 10a, b). This demonstrates that an adapted NO$^+$ CIMS instrument retains
sensitive measurement of atmospherically important species such as aromatics that are often
targeted using PTRMS and in addition can detect compounds such as iso-pentane, sum of 2- and
3-methylpentanes, methylcyclopentane, and sum of C7 cyclic alkanes (Fig. 10c-f) that are usually
not detected with PTR-MS. Slopes for calibrated VOCs, and correlation coefficients ($R^2$) for all
VOCs investigated, are included in Table 3.
To assess the ability of the NO$^+$ Tof-CIMS to separate ketones and aldehydes, we explore
measurements of propanal and acetone. The separate measurement of these two species is a good
test case because the two peaks are chromatographically well resolved on the GC-EIMS, there are
few isomers of $C_3H_6O$ (of which acetone and propanal are likely the only atmospherically relevant
species), and independent measurements of these two species are interesting for scientific reasons:
aldehydes are generally much more reactive with OH than their ketone isomers and may have
significantly different behavior in the atmosphere (Atkinson and Arey, 2003).
A time-series of propanal and acetone is shown in Fig. 11a. The two compounds have
clearly different behavior in the atmosphere: there is fast (seconds to minutes), high variability in
the acetone measurement that is not seen in the propanal measurement, and the longer term
(~hours) variability of acetone and propanal is not the same. The fast, high spikes in acetone may
come from local sources such as exhaust from chemistry labs in the building. The acetone
comparison between the GC-EIMS and the $NO^+$ ToF-CIMS has a slope of 1.13, a correlation
coefficient $R^2$ of 0.978 and negligible offset. The comparison between the GC and CIMS propanal
measurements has an $R^2$ of 0.928 (Fig. 11b, c).
Several episodes occurred with elevated high-mass n-alkane masses (m/z 169 $C_{12}H_{25}^+$,
dodecane; m/z 183 $C_{13}H_{27}^+$, tridecane; m/z 197 $C_{14}H_{29}^+$, tetradecane; m/z 211 $C_{15}H_{31}^+$,
pentadecane). Two examples are shown in Fig. 12. The episodes show high temporal and
compositional variability. The inlet was downwind from a parking lot, and next to a loading dock
and electric power generator for the building, and it is likely that the elevated C12-C15 alkanes
are from any or all of these sources. An ambient air measurement of these species is particularly
interesting because they have been implicated in efficient secondary organic aerosol production
from diesel fuel exhaust (Gentner et al., 2012).
**4. Summary and conclusions**
In summary, an $H_3O^+$ ToF-CIMS (PTR-MS) instrument was easily and inexpensively
converted into an $NO^+$ CIMS by replacing the reagent source gas and modifying the ion source
and drift tube voltages. The usefulness of $NO^+$ CIMS for atmospheric VOC measurement was then
evaluated by (1) using a GC interface to determine product ion distributions for nearly 100 VOCs
and compare the sensitivity and simplicity of spectra to $H_3O^+$ CIMS, (2) measuring ambient air
with a GC interface, to map product ions to their VOC precursors and determine which ions may
be useful for chemically specific measurement, and (3) measuring ambient air directly, to evaluate
chemical specificity and investigate fast (1Hz) time measurement of new compounds.
Additionally, the $NO^+$ CIMS response to C12-C15 n-alkanes, and to variable humidity was
determined in some detail. Further work is needed to better understand the humidity dependence.
$NO^+$ CIMS is a valuable technique for atmospheric measurement because it can separate
small carbonyl isomers, it can provide fast and chemically specific measurement of cyclic and a
few important branched alkanes (notably, isopentane and methylpentanes) that cannot be detected
by PTR-MS, it can measure alkyl-substituted aromatics with less fragmentation than $H_3O^+$ CIMS,



and it can detect larger (C12-C15) alkanes. With $NO^+$ CIMS significant fragmentation of most
small alkanes does occur, making them difficult to measure quantitatively. There are also
interferences on many alcohols (with the exception of ethanol) and butanal. Additionally, it is
worth considering that $VOC \cdot NO^+$ cluster formation moves certain species into a higher mass range.
This may be a drawback because the number of possible isobaric compounds increases with mass,
and it may be more difficult for high-resolution peak-fitting algorithms to separate species of
interest from isobaric interferences (example in Fig. S8). Finally, because there are three different
ionization mechanisms, (hydride transfer, charge transfer, and $NO^+$ adduct formation), it may be
difficult to determine which VOC precursors correspond to particular ions. $NO^+$ CIMS may be an
extremely useful supplementary approach for specific applications such as studying secondary
organic aerosol precursors in vehicle exhaust, investigating emissions from oil and natural gas
extraction, identifying additional species in complex emissions such as biomass burning,
measuring emissions of oxygenated consumer products and solvents in urban areas, and
investigating photochemistry of biogenic VOCs.
**Author Contribution**
P. Veres and C. Warneke obtained project funding. B. Yuan, A. Koss, C. Warneke, and J.
de Gouw developed the ToF-CIMS instrument. A. Koss converted the instrument from $H_3O^+$ to
$NO^+$, designed the experiments, collected data, and wrote the manuscript. A. Koss and M. Coggon
analyzed data. C. Warneke and J. de Gouw provided guidance on experimental design and
interpretation. All authors edited the manuscript.
**Acknowledgements**
This work was funded by the CIRES Innovative Research Program. A. R. Koss
acknowledges additional support from the NSF Graduate Fellowship Program. We would like to
thank J. B. Gilman and B. M. Lerner for help with GC operation and data analysis.

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



**Tables.**

Table 1. VOC species in Fig. 4 and their charge transfer and hydride transfer reaction enthalpies.

| ID # | Species name | Hydride transfer enthalpy (kJ/mol) | Charge transfer enthalpy (kJ/mol) |
|---|---|---|---|
| 0 | methanol | 22.98 | 152.05 |
| 1 | ethene | 174.58 | 120.59 |
| 2 | acetaldehyde | -61.32 | 93.20 |
| 3 | ethane | 100.98 | 217.65 |
| 4 | ethanol | -68.02 | 117.31 |
| 5 | propene | 40.17 | 44.96 |
| 6 | propanal | -105.32 | 67.15 |
| 7 | propane | 8.88 | 161.69 |
| 8 | n-propanol | -78.72 | 92.23 |
| 9 | i-propanol | -122.22 | 87.41 |
| 10 | methacrolein | -87.62 | 63.29 |
| 11 | 1-butene | -39.39 | 27.59 |
| 12 | iso-butene | 15.88 | -4.24 |
| 13 | 2-butenes | 17.18 | -15.82 |
| 14 | butanal | -84.02 | 53.64 |
| 15 | n-butane | 6.98 | 122.14 |
| 16 | iso-butane | -56.32 | 136.61 |
| 17 | 1-butanol | -87.02 | 70.04 |
| 18 | 2-methylpropanol | -94.02 | 72.94 |
| 19 | 2-butanol | -137.02 | 59.43 |
| 20 | 1,4-pentadiene | -69.32 | 34.35 |
| 21 | 1-pentene | -53.02 | 21.80 |
| 22 | 2-pentene | -92.02 | -23.54 |
| 23 | 3-methyl-1-butene | -92.52 | 24.70 |
| 24 | cyclopentane | -7.22 | 102.84 |
| 25 | n-pentane | -6.02 | 98.02 |
| 26 | iso-pentane | -70.02 | 101.88 |
| 27 | neo-pentane | 77.98 | 99.95 |
| 28 | 1-pentanol | -94.02 | 97.05 |
| 29 | 3-methyl-2-butanol | -143.02 | 51.71 |
| 30 | 3-pentanol | -140.02 | 47.85 |
| 31 | benzene | 159.08 | -1.93 |
| 32 | cyclohexane | -28.02 | 59.43 |
| 33 | methylcyclopentane | -80.02 | 42.06 |
| 34 | 4-methyl-2-pentene | -117.82 | -27.40 |
| 35 | 3-methyl-1-pentene | -125.22 | 16.98 |
| 36 | 2,3,-dimethyl-1-butene | -94.02 | -18.72 |
| 37 | n-hexane | -13.92 | 83.55 |
| 38 | 2-methylpentane | -74.72 | 72.94 |
| 39 | 2,3-dimethylbutane | -79.22 | 66.18 |
| 40 | 3-methylpentane | -75.42 | 69.08 |
| 41 | toluene | -36.02 | -42.06 |
| 42 | methylcyclohexane | -73.02 | 36.27 |
| 43 | 1,2-dimethyl-cyclopentane | -95.52 | 63.29 |
| 44 | ethylbenzene | -103.02 | -47.66 |
| 45 | o-xylene | -55.02 | -67.92 |
| 46 | m-xylene | -47.22 | -68.88 |
| 47 | p-xylene | -65.92 | -79.50 |
| 48 | isopropylbenzene | -111.92 | -51.52 |
| 49 | 3-ethyltoluene | -103.12 | -82.41 |
| 50 | acetone | | 43.03 |
| 51 | butanone | | 24.70 |
| 52 | 2-pentanone | | 11.19 |
| 53 | 3-pentanone | | 4.44 |
| 54 | MVK | | 37.24 |





Atmospheric Measurement Techniques Discussions — Open Access — EGU

**Table 2.** Sensitivities and detection limits of NO$^+$ ToF-CIMS for various VOCs. Additional product ions are listed in gray text.

**A. Species calibrated directly with NO$^+$ CIMS**

| VOC species | Ion formula (% of total signal) — Formula | Mech-anism | (% of total signal) | Exact m/z (Th) | Back-ground cps | Noise scale factor α | NO$^+$ sensitivity ncps/ppb | NO$^+$ sensitivity cps/ppb | NO$^+$ 1-s detection limit | H$_3$O$^+$ CIMS 1s detection limit |
|---|---|---|---|---|---|---|---|---|---|---|
| Methanol | CH$_4$ONO$^+$ | M+NO$^+$ | (12%) | 62.024 | 0.88 | 1.23 | 0.15 | 0.45 | 28 ppb | 0.397 ppb |
| | CH$_4$OH$^{+*}$ | M+H$^+$ | (49%) | 33.034 | | | | | | |
| | CH$_7$O$_2^{+*}$ | M+H$_3$O$^+$ | (39%) | 51.044 | | | | | | |
| Acetonitrile | C$_2$H$_3$NNO$^+$ | M+NO$^+$ | (48%) | 71.024 | 1.5 | 1.33 | 7 | 30 | 503 ppt | 45 ppt |
| | C$_2$H$_3$NH$^{+*}$ | M+H$^+$ | (44%) | 42.034 | | | | | | |
| | C$_2$H$_6$NO$^{+*}$ | M+H$_3$O$^+$ | (8%) | 60.044 | | | | | | |
| Acetaldehyde | C$_2$H$_3$O$^+$ | M-H$^-$ | (60%) | 43.018 | 46 | 1.33 | 41 | 133 | 337 ppt | 195 ppt |
| | C$_2$H$_5$O$_2^+$ | M-H+H$_2$O | (13%) | 61.028 | | | | | | |
| | C$_2$H$_4$OH$^{+*}$ | M+H$^+$ | (11%) | 45.034 | | | | | | |
| | C$_2$H$_4$ONO$^+$ | M+NO$^+$ | (9%) | 74.024 | | | | | | |
| Acetone | C$_3$H$_6$ONO$^+$ | M+NO$^+$ | (82%) | 88.039 | 28 | 1.16 | 86 | 394 | 80 ppt | 97 ppt |
| | C$_3$H$_6$OH$^{+*}$ | M+H$^+$ | (13%) | 59.049 | | | | | | |
| Isoprene | C$_5$H$_8^+$ | M$^+$ | (46%) | 68.062 | 0.93 | 1.34 | 59 | 242 | 58 ppt | 162 ppt |
| | C$_5$H$_8$NO$^+$ | M+NO$^+$ | (17%) | 98.060 | | | | | | |
| | C$_5$H$_7^+$ | M-H$^-$ | (7%) | 67.054 | | | | | | |
| MEK | C$_4$H$_8$ONO$^+$ | M+NO$^+$ | (86%) | 102.055 | 4.1 | 1.33 | 157 | 781 | 24 ppt | 45 ppt |
| | C$_4$H$_8$OH$^{+*}$ | M+H$^+$ | (8%) | 73.065 | | | | | | |
| Benzene† | C$_6$H$_6^+$ | M$^+$ | (55%) | 78.046 | 11 | 1.37 | 68 | 302 | 88 ppt | |
| | C$_6$H$_6$NO$^+$ | M+NO$^+$ | (40%) | 108.044 | 10 | 1.72 | 50 | 254 | 123 ppt | 96 ppt |
| | *sum* | | | | *21* | *1.59* | *116* | *556* | *69 ppt* | |
| Toluene | C$_7$H$_8^+$ | M$^+$ | (89%) | 92.062 | 19 | 1.33 | 138 | 663 | 47 ppt | 47 ppt |
| | C$_7$H$_8$NO$^+$ | M+NO$^+$ | (8%) | 122.060 | | | | | | |
| o-Xylene | C$_8$H$_{10}^+$ | M$^+$ | (94%) | 106.078 | 4.2 | 1.51 | 154 | 789 | 28 ppt | 40 ppt |
| | C$_8$H$_{10}$NO$^+$ | M+NO$^+$ | (5%) | 136.076 | | | | | | |
| 1,2,4-Trimethylbenzene | C$_9$H$_{12}^+$ | M$^+$ | (100%) | 120.093 | 1.3 | 1.75 | 162 | 882 | 22 ppt | 45 ppt |
| n-Pentadecane | C$_{15}$H$_{31}^+$ | M-H$^-$ | (72%) | 211.242 | 2.7 | 1.83 | 48 | 512 | 46 ppt | --- |
| | C$_9$H$_{19}^+$ | fragment | (3%) | 127.148 | | | | | | |
| | C$_{10}$H$_{21}^+$ | fragment | (3%) | 141.164 | | | | | | |
| | C$_8$H$_{17}^+$ | fragment | (3%) | 113.132 | | | | | | |

**B. Sensitivity estimated via sensitivity relative to H$_3$O$^+$ CIMS**

| VOC species | H$_3$O$^+$ cps/ ppb | Product ions — Formula | Mech-anism | (% of total signal) | Exact m/z (Th) | Relative (NO$^+$cps/ H$_3$O$^+$cps) | Back-ground cps | Noise scale factor α | NO$^+$ sensitivity ncps/ ppb | NO$^+$ sensitivity cps/ ppb | NO$^+$ 1s detection limit | H$_3$O$^+$ CIMS 1s detection limit |
|---|---|---|---|---|---|---|---|---|---|---|---|---|
| Ethanol | 119 | C$_2$H$_5$O$^+$ | M-H$^-$ | (80%) | 45.033 | 6.2 | 149 | 1.37 | 127 | 738 | 105 ppt | 1627 ppt |
| | | C$_2$H$_7$O$_2^+$ | M-H+H$_2$O | (15%) | 63.044 | | | | | | | |
| | 27 | C$_7$H$_{13}^+$ | M-H$^-$ | (98%) | 97.101 | 17 | 6.6 | 1.32 | 53 | 448 | 50 ppt | 943 ppt |




| | | | | | | | | | | | | |
|---|---|---|---|---|---|---|---|---|---|---|---|---|
| Methyl-cyclohexane | | $C_6H_{11}^+$ | fragment | (2%) | 83.086 | | | | | | | |
| MVK | 539 | $C_4H_6ONO^+$ | $M+NO^+$ | (100%) | 100.039 | 0.38 | 4 | 1.71 | 24 | 202 | 112 ppt | 85 ppt |
| Pentanones | 770 | $C_5H_{10}ONO^+$ | $M+NO^+$ | (83%) | 116.071 | 1.18 | 4.4 | 1.32 | 97 | 906 | 21 ppt | 47 ppt |
| | | $C_5H_{10}OH^{+}$* | $M+H^+$ | (7%) | 87.080 | | | | | | | |
| α-Pinene | 262 | $C_{10}H_{16}^+$ | $M^+$ | (59%) | 136.125 | 0.28 | 0.39 | 1.69 | 7.3 | 73 | 233 ppt | 67 ppt |
| | | $C_7H_8^+$ | fragment | (24%) | 92.062 | | | | | | | |
| | | $C_7H_9^+$ | fragment | (11%) | 93.070 | | | | | | | |
| | | $C_{10}H_{16}H^{+}$* | $M+H^+$ | (7%) | 137.132 | | | | | | | |

**C. Sensitivity estimated via correlation with GC-EIMS**

| VOC species | Formula | Mechanism | Product ions (% of total signal) | Exact m/z (Th) | Correlation with GC ($R^2$) | Back-ground cps | Noise scale factor α | $NO^+$ Sensitivity ncps/ppb | cps/ppb | $NO^+$ 1-s detection limit |
|---|---|---|---|---|---|---|---|---|---|---|
| Propanal | $C_3H_5O^+$ | $M-H^-$ | (65%) | 57.033 | 0.928 | 11 | 1.40 | 170 | 1057 | 26 ppt |
| | $C_3H_7O_2^{+}$* | $M-H+H_2O$ | (17%) | 75.044 | | | | | | |
| | $C_3H_6OH^{+}$* | $M+H^+$ | (7%) | 59.049 | | | | | | |
| Methacrolein + crotonaldehyde | $C_4H_5O^+$ | $M-H^-$ | (64%) | 69.033 | 0.984 | 4.1 | 1.37 | 48 | 325 | 60 ppt |
| | $C_4H_6ONO^+$ | $M+NO^+$ | (16%) | 100.039 | | | | | | |
| | $C_3H_5^+$ | fragment | (10%) | 41.039 | | | | | | |
| iso-Pentane | $C_5H_{11}^+$ | $M-H^-$ | (82%) | 71.086 | 0.888 | 23 | 1.36 | 101 | 706 | 49 ppt |
| | $C_3H_7^+$ | fragment | (11%) | 43.054 | | | | | | |
| Methylcyclopentane | $C_6H_{11}^+$ | $M-H^-$ | (99%) | 83.086 | 0.961 | 7.4 | 1.34 | 154 | 1225 | 18 ppt |
| C5 aldehydes | $C_5H_9O^+$ | $M-H^-$ | (49%) | 85.065 | 0.936 | 9.8 | 1.38 | 119 | 904 | 28 ppt |
| | $C_4H_9^+$ | fragment | (22%) | 57.070 | | | | | | |
| | $C_5H_{11}O_2^{+}$* | $M-H+H_2O$ | (19%) | 103.075 | | | | | | |
| 2- and 3-methylpentane | $C_6H_{13}^+$ | $M-H^-$ | (82%) | 85.101 | 0.978 | 16 | 1.34 | 122 | 981 | 30 ppt |
| | $C_3H_7^+$ | fragment | (10%) | 43.054 | | | | | | |
| | $C_4H_9^+$ | fragment | (4%) | 57.070 | | | | | | |
| Hexanal | $C_6H_{11}O^+$ | $M-H^-$ | (49%) | 99.080 | 0.945 | 10 | 1.47 | 160 | 1270 | 22 ppt |
| | $C_6H_{13}O_2^{+}$* | $M-H+H_2O$ | (23%) | 117.091 | | | | | | |
| | $C_5H_{11}^+$ | fragment | (15%) | 71.086 | | | | | | |
| Styrene | $C_8H_8^+$ | $M^+$ | (100%) | 104.062 | 0.949 | 0.62 | 1.47 | 112 | 966 | 15 ppt |
| Benzaldehyde | $C_7H_5O^+$ | $M-H^-$ | (100%) | 105.033 | 0.923 | 12 | 1.37 | 75 | 621 | 43 ppt |

\* Product from residual $H_3O^+$

† Both product ions can be unambiguously assigned to benzene. We therefore report also the counting statistics and limit of detection for the sum of the two ions.



**Table 3**. Assessment of significant product ions investigated by GC-NO$^+$ CIMS and parallel GC-EIMS and NO$^+$ CIMS measurement of ambient air. Masses in bold can be unambiguously assigned to a single VOC or a structurally related, correlated group of VOCs.

| Ion formula | Exact mass (Th) | Assessment from series GC-NO$^+$ ToF-CIMS | Correlation with parallel GC-EIMS R$^2$ | Slope (ppbv/ppbv) |
|---|---|---|---|---|
| C$_3$H$_5$$^+$ | 41.039 | several non-correlated species | | |
| **C$_2$H$_3$O$^+$** | 43.018 | **acetaldehyde** | **0.942** | **0.892** |
| C$_3$H$_7$$^+$ | 43.054 | several non-correlated species | | |
| **C$_2$H$_5$O$^+$** | 45.033 | **ethanol** | **0.998** | |
| **C$_4$H$_6$$^+$** | **54.046** | **propyne[1]** | | |
| C$_4$H$_8$$^+$ | 56.062 | several non-correlated species | | |
| **C$_3$H$_5$O$^+$** | 57.033 | **propanal** | **0.928** | |
| C$_4$H$_9$$^+$ | 57.070 | several non-correlated species | | |
| C$_3$H$_7$O$^+$ | 59.049 | interference from acetone; if accounted for, sum of C3 alcohols | | |
| **CH$_4$NO$_2$$^+$** | 62.024 | **methanol, but poor sensitivity** | **0.904** | **1.25** |
| C$_5$H$_6$$^+$ | 66.046 | interference from benzene; if accounted for, cyclopentadiene | | |
| **C$_4$H$_4$O$^+$** | **68.026** | **furan[2]** | | |
| C$_5$H$_8$$^+$ | 68.062 | possibly: isoprene[3] | | |
| **C$_4$H$_5$O$^+$** | **69.033** | **methacrolein + crotonaldehyde[4]** | **0.984** | |
| C$_5$H$_9$$^+$ | 69.070 | several non-correlated species | | |
| C$_5$H$_{10}$$^+$ | 70.078 | possibly: sum of 2-pentenes[3] | | |
| C$_4$H$_7$O$^+$ | 71.049 | several non-correlated species | | |
| **C$_5$H$_{11}$$^+$** | **71.086** | **iso-pentane** | **0.888** | |
| C$_4$H$_9$O$^+$ | 73.065 | several non-correlated species | | |
| **C$_6$H$_6$$^+$** | **78.046** | **benzene[5]** | **0.987** | **0.847** |
| C$_5$H$_6$O$^+$ | 82.041 | possibly: sum of 2- and 3-methylfuran[3] | | |
| **C$_6$H$_{11}$$^+$** | **83.086** | **methylcyclopentane** | **0.961** | |
| **C$_5$H$_9$O$^+$** | **85.065** | **sum of C5 aldehydes** | **0.936** | |
| **C$_6$H$_{13}$$^+$** | **85.101** | **sum of 2- and 3-methylpentane** | **0.978** | |
| C$_4$H$_8$NO$^+$ | 86.060 | several non-correlated species | | |
| C$_5$H$_{11}$O$^+$ | 87.080 | C5 alcohols and ethers; significant interference from minor carbonyl product ions | | |
| **C$_3$H$_6$NO$_2$$^+$** | **88.039** | **acetone** | **0.978** | **1.13** |
| C$_2$H$_4$NO$_3$$^+$ | 90.019 | possibly: acetic acid (chromatography too poor to determine) | | |
| **C$_7$H$_8$$^+$** | **92.062** | **toluene** | **0.999** | **0.810** |
| **C$_7$H$_{13}$$^+$** | **97.101** | **sum of C7 cyclic alkanes** | **0.917** | |
| **C$_6$H$_{11}$O$^+$** | **99.080** | **hexanal** | **0.945** | |
| C$_7$H$_{15}$$^+$ | 99.117 | possibly: sum of 2- and 3-methylhexane, but poor sensitivity | | |
| **C$_4$H$_6$NO$_2$$^+$** | **100.039** | **MVK** | **0.950** | |
| C$_5$H$_{10}$NO$^+$ | 100.076 | possibly: sum of C5 terminal alkenes, but poor sensitivity | | |
| **C$_4$H$_8$NO$_2$$^+$** | **102.055** | **MEK** | **0.971** | **0.843** |
| **C$_8$H$_8$$^+$** | **104.062** | **styrene (vinyl benzene)** | **0.949** | |
| **C$_7$H$_5$O$^+$** | **105.033** | **benzaldehyde** | **0.923** | |
| **C$_8$H$_{10}$$^+$** | **106.078** | **sum of C8 aromatics** | **0.952** | **0.746** |
| **C$_6$H$_6$NO$^+$** | **108.044** | **benzene[5]** | | |
| C$_8$H$_{15}$$^+$ | 111.117 | possibly: sum of C2 alkyl-substituted cyclohexanes[6] | 0.761 | |
| **C$_7$H$_{13}$O$^+$** | **113.096** | **heptanal[2]** | | |
| C$_8$H$_{17}$$^+$ | 113.132 | possibly: sum of methylheptanes, but poor sensitivity | | |
| **C$_5$H$_{10}$NO$_2$$^+$** | **116.071** | **sum of C5 ketones** | **0.945** | |
| C$_9$H$_{10}$$^+$ | 118.078 | possibly: sum of methylstyrene isomers[3] | | |
| C$_9$H$_{12}$$^+$ | 120.093 | sum of C9 aromatics; scatter possibly due to disparity in response factors | 0.600 | |
| **C$_8$H$_{15}$O$^+$** | **127.112** | **octanal[2]** | | |





| | | | |
|---|---|---|---|
| $C_6H_{12}NO_2^+$ | 130.086 | possibly: sum of C6 ketones[3] | |
| $C_{10}H_{14}^+$ | 134.109 | possibly: sum of C10 aromatics | |
| $C_{10}H_{16}^+$ | 136.125 | monoterpenes plus unknown interference; possibly adamantane from vehicle exhaust | 0.584 |
| $C_7H_{14}NO_2^+$ | 144.102 | heptanone[2] | |

[1] Cross-comparison with independent GC-EIMS not possible due to chromatographic quantitation ion overlap with neighboring peaks.

[2] Cross-comparison with independent GC-EIMS not possible due to EIMS quadrupole SIS (selected ion scan) window restrictions.

[3] Concentrations too low in ambient air to determine.

[4] Winter urban air sampled was likely influenced by local domestic biomass burning; crotonaldehyde may be a smaller fraction of signal in other environments.

[5] Benzene correlation using sum of m108 $C_6H_6NO^+$ and m78 $C_6H_6^+$.

[6] With exclusion of single outlier, $R^2 = 0.831$.




## Figures

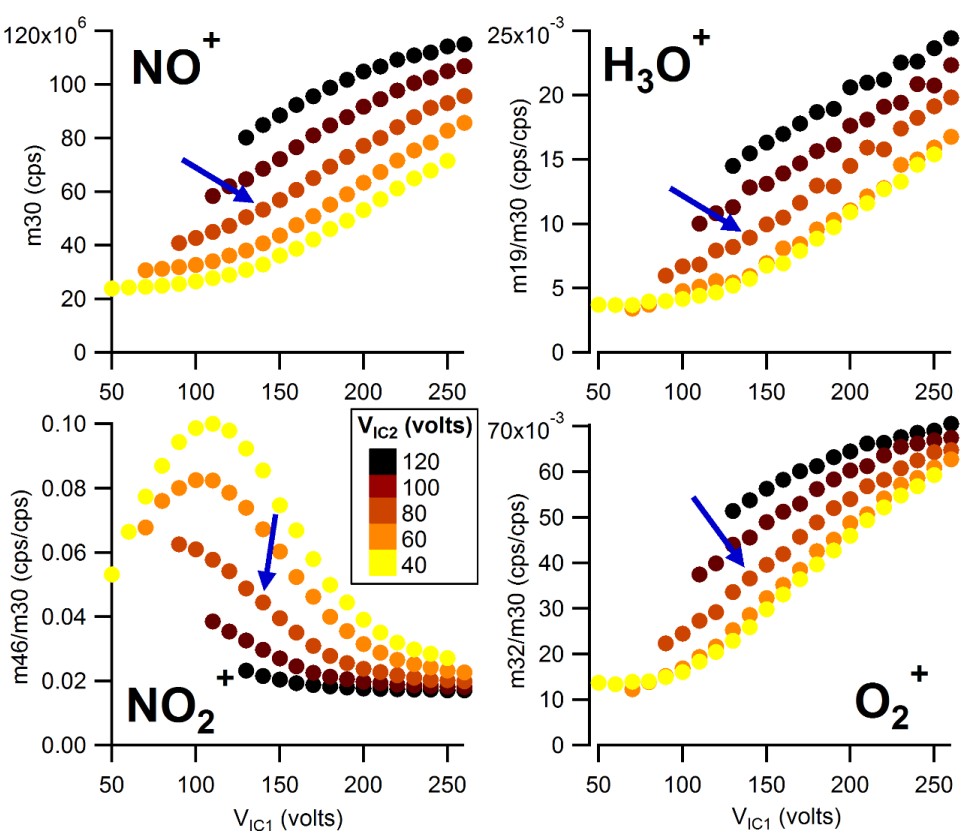

Figure 1. Dependence of $NO^+$, $H_3O^+$, $NO_2^+$, and $O_2^+$ on intermediate chamber voltages. The arrow denotes the selected operating conditions.



Figure 2. VOC and primary product ion dependence on drift tube voltage. Traces are labeled by the nominal product ion m/z in Th. (a) Methyl vinyl ketone. (b) Methacrolein. (c) 2,2-dimethylbutane. (d) Methylcyclohexane. (e) Primary ions and clusters. The dashed line indicates the selected operating voltage.



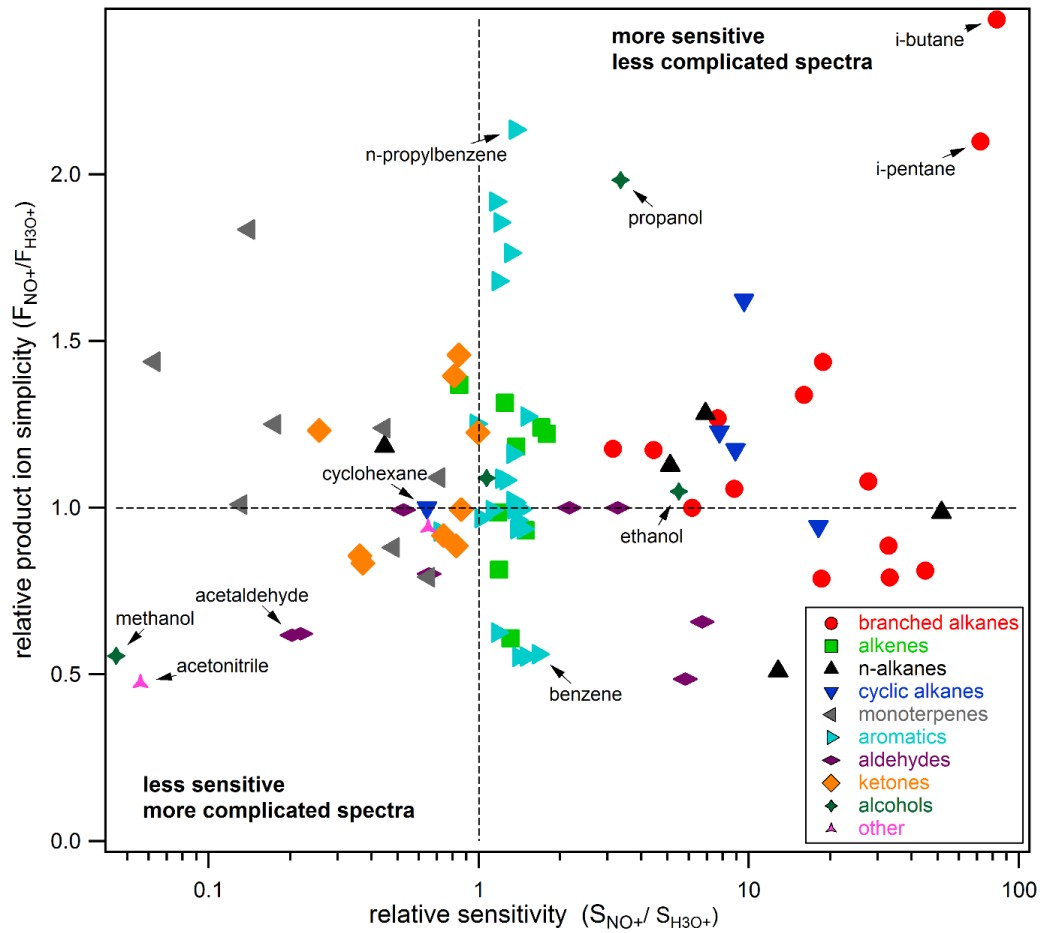

Figure 3. Comparison of production ion distribution and sensitivity of VOCs using NO$^+$ and H$_3$O$^+$ reagent ion chemistry.





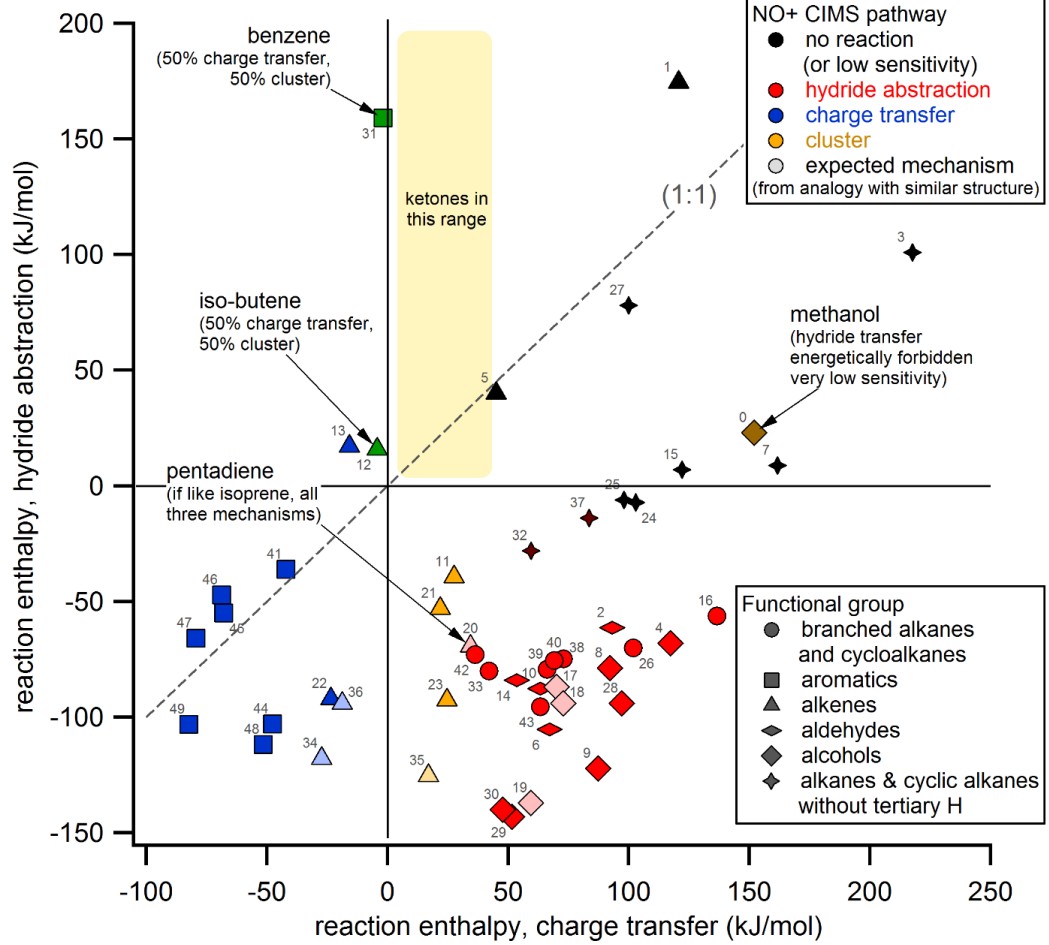

Figure 4. VOC-NO$^+$ reaction mechanism dependence on charge transfer and hydride transfer reaction enthalpy. VOC identification is indicated by the small numbers and is listed in Table 1. Hydride abstraction enthalpies for ketones are not known, but can be assumed to be positive based on structural considerations (lack of tertiary hydrogen). Ion thermodynamic information is available for several species whose reaction mechanism was not experimentally verified in this work; an expected mechanism was determined by analogy with a VOC of similar structure:

**17** 1-butanol; by analogy with 1-propanol.

**18** 2-methylpropanol; by analogy with 1-propanol.

**19** 2-butanol; by analogy with 2-propanol.

**20** 1,4-pentadiene; by analogy with isoprene.

**34** 4-methyl-2-pentene; by analogy with 2-pentene.

**35** 3-methyl-1-pentene; by analogy with 1-hexene.

**36** 2,3,-dimethyl-1-butene; by analogy with iso-butene.





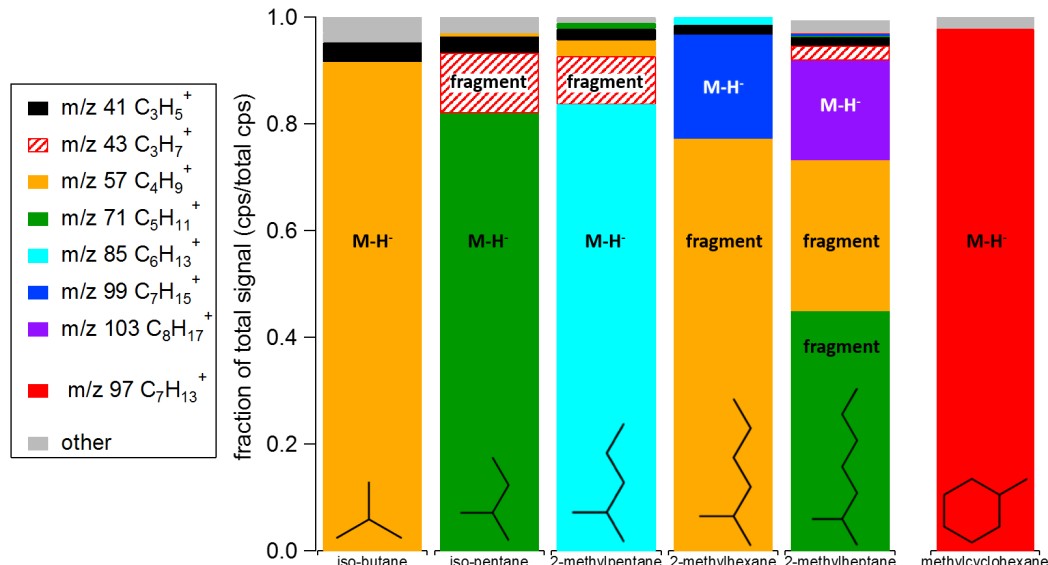

Figure 5. Product ion distributions of selected aliphatic hydrocarbons.




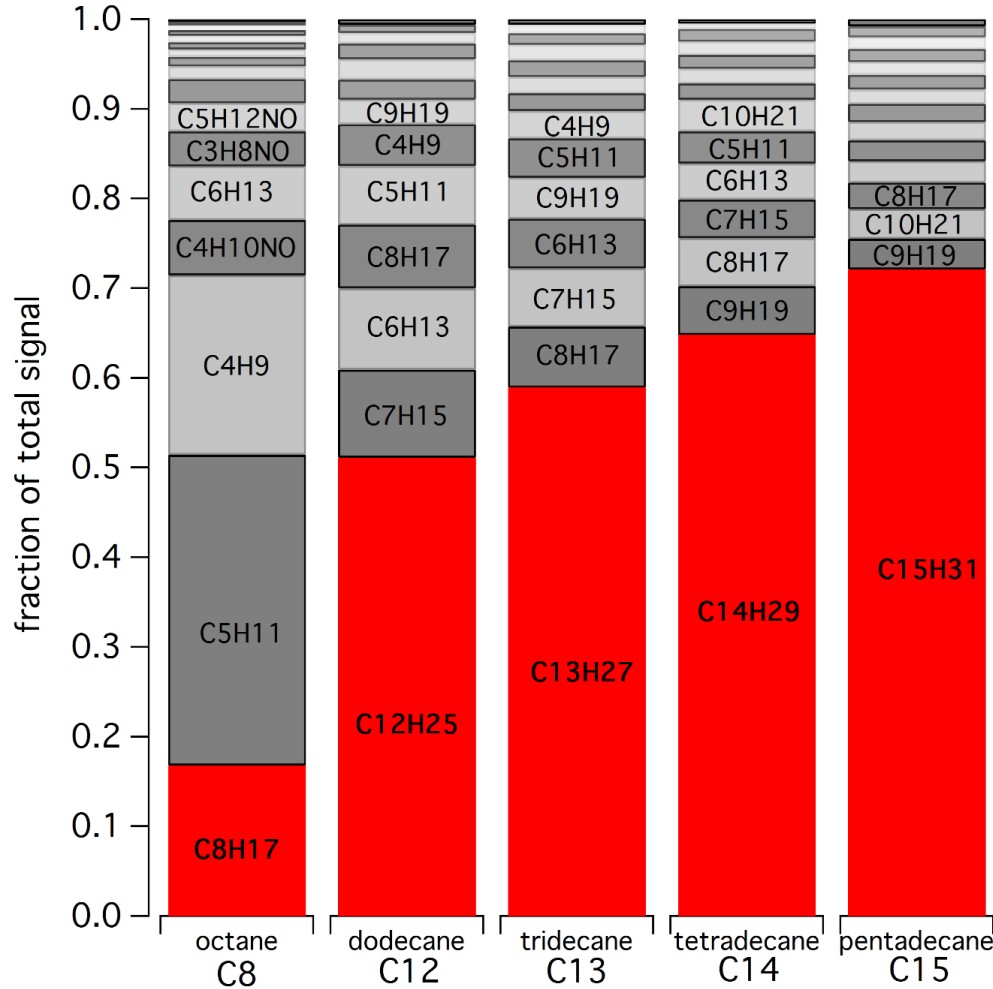

Figure 6. Large (C12-C15) n-alkane product ion distribution. The expected largest mass resulting from hydride abstraction (*m-1*) is highlighted in red. N-octane (C8) is shown for comparison.



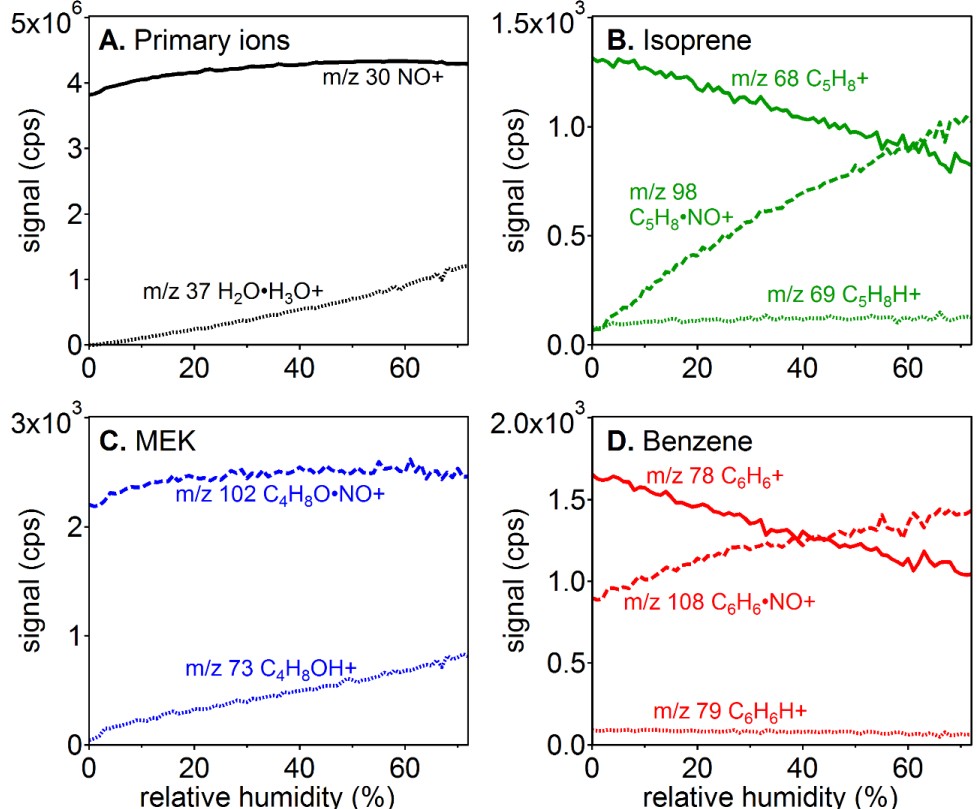

Figure 7. Humidity dependence of primary ions and selected VOCs. (a) NO$^+$ and water clusters. (b) isoprene. (c) methyl ethyl ketone (MEK). (d) benzene.





Figure 8. Example GC-CIMS chromatogram of ambient air sample. Masses have been split between two panels for clarity. Top: select masses corresponding to branched and cyclic alkanes. Bottom: select masses corresponding to aldehydes and ketones.





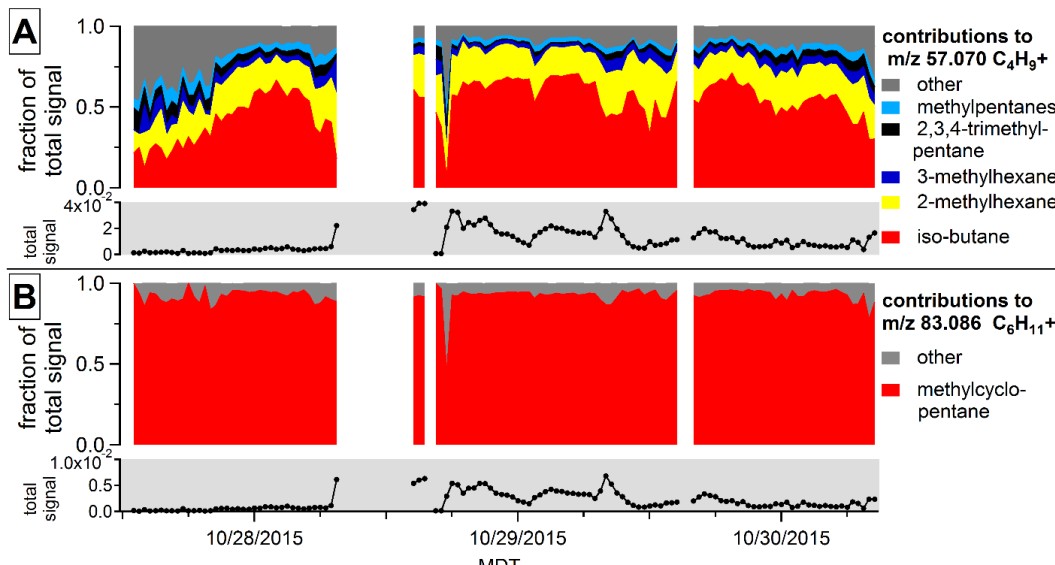

Figure 9. Contributions to two masses based on GC-CIMS measurements of ambient air. "Total signal" is normalized counts per chromatogram. (a) m/z 57 $C_4H_9^+$. (b) m/z 83 $C_6H_{11}^+$.

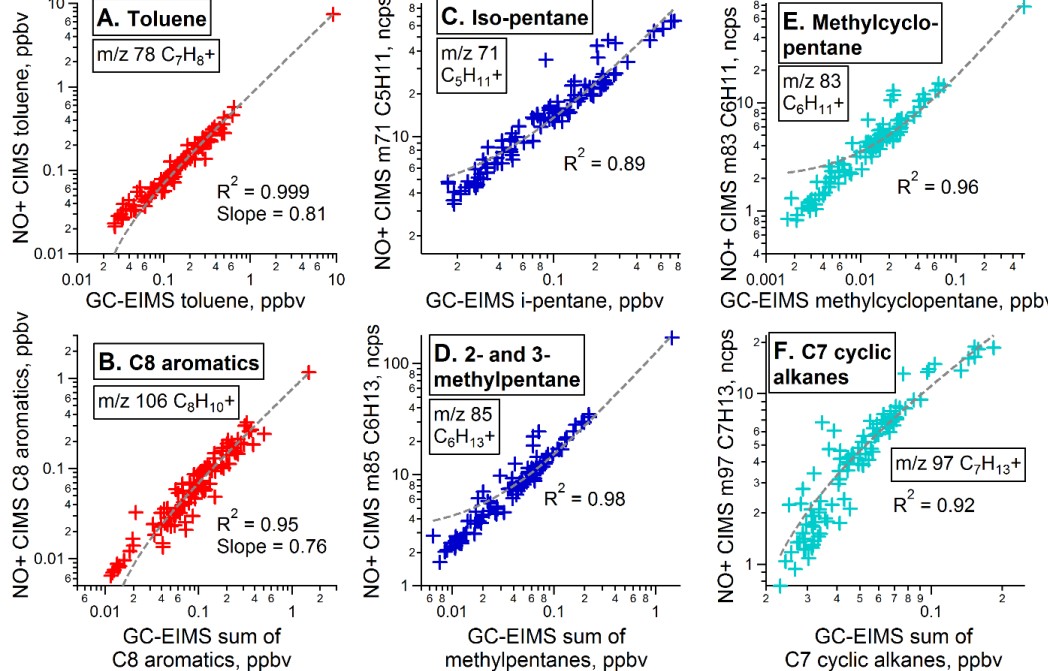

Figure 10. Correlations between VOCs measured with GC-EIMS and $NO^+$ ToF-CIMS. The 1Hz $NO^+$ ToF-CIMS measurement is averaged to the 5 minute GC collection period. Orthogonal least-squares linear best fits (ODR best fit) are shown with dashed lines. The lines appear curved due to log scale axes. For several compounds (e.g. methylcyclopentane, 2-and 3 methylpentanes), the




single high outlier pulls the best fit slightly away from the data points at low mixing ratios. (a) Toluene. (b) C8 aromatics: sum of ethylbenzene, o-xylene, m-xylene, and p-xylene. (c) Iso-pentane. (d) Sum of 2-methylpentane and 3-methylpentane. (e) Methylcyclopentane. (f) C7 cyclic alkanes: sum of methylcyclohexane, ethylcyclopentane, and dimethylcyclopentanes.

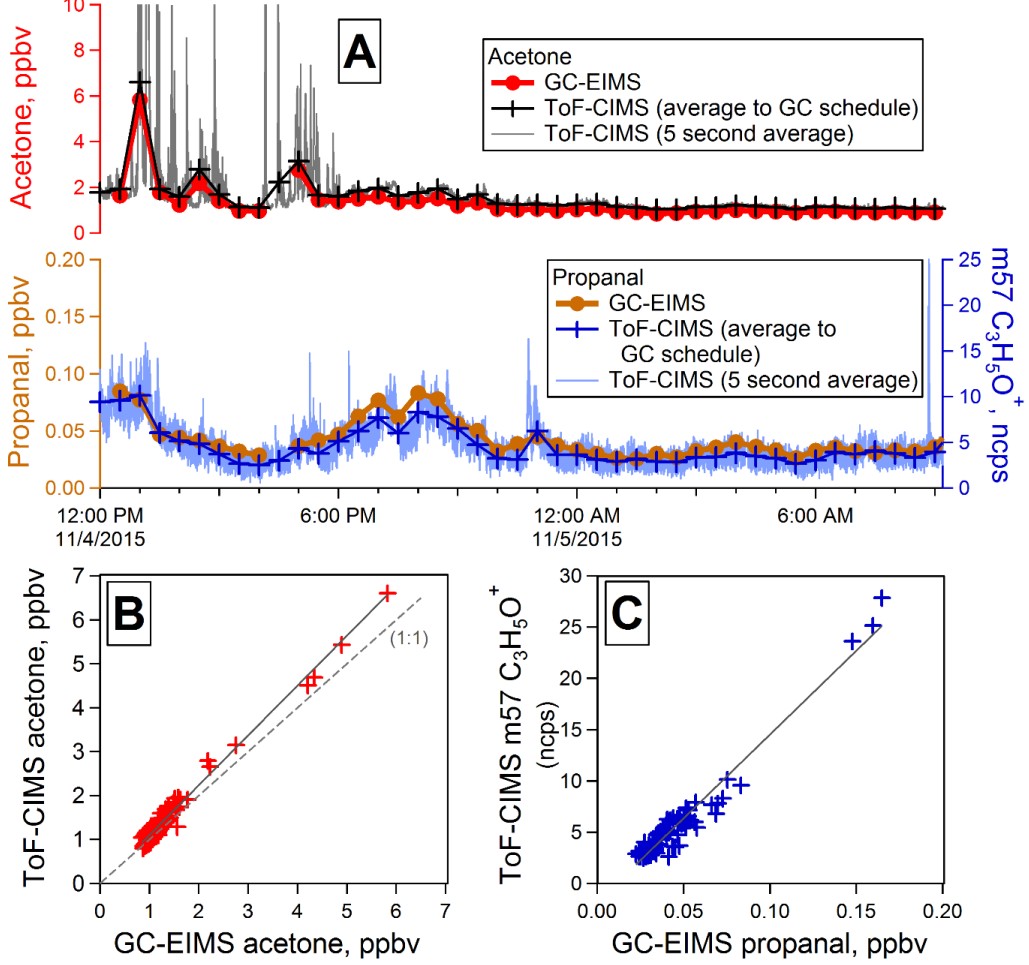

Figure 11. (a) Time series of acetone and propanal measurements from NO+ ToF-CIMS and GC-EIMS. Measurements shown include the GC-EIMS measurement (5 minute sample every 30 minutes, circle markers), the NO[+] ToF-CIMS measurement averaged over the five-minute GC sampling period (cross markers), and the NO+ ToF-CIMS measurement averaged to a 5 second running mean. (b) Correlation between NO[+] ToF-CIMS and GC-EIMS measurement of acetone. (c) Correlation between NO[+] ToF-CIMS and GC-EIMS measurement of propanal.



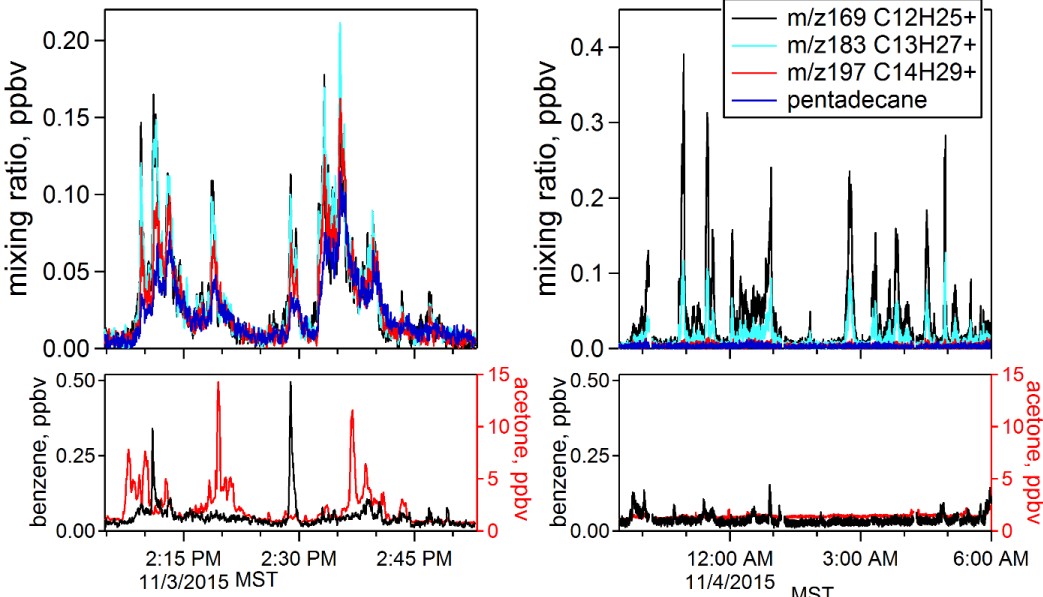

Figure 12. Episodes with elevated high-mass alkane masses. Mixing ratios for m/z 169 $C_{12}H_{25}^+$ (dodecane), m/z 183 $C_{13}H_{27}^+$ (tridecane), and m/z 197 $C_{14}H_{29}^+$ (tetradecane) are shown in approximate ppbv, assuming the same instrument calibration factor as pentadecane. Additional VOC species (benzene, acetone) are shown in the bottom panels for context.