# Peer review of "Evaluation of NO+ reagent ion chemistry for on-line measurements of atmospheric volatile organic compounds"

_Atmospheric Measurement Techniques, 2016_

## Referee Comment (RC1) · Anonymous Referee #3 · 5 May 2016

General comments:

This manuscript presents a fairly straight-forward way to adapt a PTR-TOF mass spectrometer to use NO+ chemical ionization. It presents a comprehensive ensemble of experiments to establish the involved ion chemistry, and the methods employed. It also offers a thorough discussion of the results and the performance obtained, in particular in comparison to widely used PTR instrumentation. Furthermore, the results are very clearly presented (although I suspect there is still a bit of room for improvement). Therefore, I think that the manuscript is very well within the scope of AMT, and I believe it will be of high interest to the chemical ionization mass spec community. I recommend its publication, with a few minor revisions or clarifications.

[Figure]

Specific comments:

Section 2 (e.g. lines 139+): How were the quadrupole ion guides set? As the authors state, these ion guides can change the measured ion distributions, e.g. via changing the m/z-dependent transmission, and adduct products can be more or less strongly de-clustered, directly affecting the presented results. (If one so desired, spectra could indeed be altered quite drastically.) So I wonder, was there any tuning done on the ion guidance elements, specifically for the presented experiments? Is it possible to compare the used quad settings with other conditions tested by the authors, or by others?

Table 2: Not totally clear to me, what the gray text means. The table caption says that gray is for additional product ions. "Additional" apparently meaning that they were not used to establish an NO+ sensitivity. I assume the last column texts are gray as well, because those detection limits do not directly relate to any of the numbers or formulas to the left. Or is there any relation to the other gray text? Could that be somehow presented in a clearer manner?

Sections 2 and 3.1.1-3.1.4 suggest that sensitivities (and simplicities) as well as reaction and fragmentation patterns were determined at dry conditions (I think that could be stated more clearly), whereas section 3.1.5 discusses humidity dependence. There seem to be some inconsistencies comparing some Figures and Tables. For instance taking MVK: Table 2B lists the product ions to be 100% C4H6ONO+ (m/z 100), but Fig. 2A, at operating voltages, shows that the product ion at m/z 100 (presumably C4H6ONO+) constitutes only ∼70% of product ions. Other example, isoprene: Table 2A lists 46% of the signal as C5H8+, 17% as C5H8NO+, 7% as C5H7+, whereas Fig. 7B, at dry conditions, suggest >90% of the signal is C5H8+, <10% from C5H8NO+ and C5H9+ (not listed in Table 2, while C5H7+ is not shown in Fig. 7B). There is a similar discrepancy for benzene, where Table 2A doesn't agree with Fig. 7D, except maybe at about 20% RH... I may have gotten confused with the differences in experimental setup/conditions, or I understand Table 2 wrong? Either way, some clarification would

be helpful, at least for this reviewer. Maybe by an additional table that summarizes conditions for each figure's/table's experiments?

Line 332: How long was a "measurement period"? And Section 3.2 in general: How did the zero measurements look like? Was the frequency of zero measurements sufficient? The frequency was once every hour for 3.2.2, but could not find out for 3.2.1 (see comment on line 332). I'd assume it was enough, but maybe show in supplement, by a figure, or by giving a few numbers.

Fig. S8 gives a glimpse at the challenges of TOF data analysis, in this case overlapping peaks, requiring "high-resolution peak-fitting algorithms" (line 422). I was missing some short statement on what software was used to tackle these and other challenges.

Line 141: "does not have that issue as strongly" is a bit too vague. I would prefer a more quantitative statement, or a suitable reference.

Line 354: I just don't get the message of this sentence into my head. I suggest the authors break it down or reformulate.

Technical corrections:

Line 11: "fast (sub 1-Hz)" ... To my understanding "sub 1-Hz" suggests slower than 1 Hz. Is "sub 1-Hz" what is really meant here, i.e. "up to 1 Hz"? Or the opposite, i.e. "1 Hz and faster"? The paper does present experiments at 1Hz and slower, but I would think faster (>1Hz) measurements would be possible as well.

---

## Short Comment (SC1) · 11 May 2016

Our group has reported product ions and their relative intensities of C3-C13 n-alkanes and C4-C10 iso-alkanes measured by NO+ CIMS (see Table A1 of the Supplementary Data in Yamada et al. (2015)). The present results are generally similar to ours. But it seems that the ratios of fragment ions to [M-H]+ ions are larger in the present study than ours. I think that the strength of the electric field of the drift tube (i.e., E/N ratio) cannot be the reason of this difference because the ratios are similar in both the studies (60 Td in the present study and 67 Td in our study). We also showed that O2+ ionization of alkanes produces the same fragment ions as NO+ ionization (see Table A1 of the Supplementary Data in Yamada et al. (2015)). Therefore, we subtracted the con-

tribution of O2+ ionization from ion signals in order to report the detection sensitivities of alkanes by NO+ CIMS. Did the authors consider the contribution of O2+ ionization when they measured alkanes by NO+ CIMS?

Yamada, H., Inomata, S., and Tanimoto H.: Evaporative emissions in three-day diurnal breathing loss tests on passenger cars for the Japanese market, Atmos. Environ., 107, 166-173, 2015.
* * *

---

## Referee Comment (RC2) · Anonymous Referee #1 · 2 Jun 2016

This is a well-written and detailed description of the conversion of an H3O+ ToF-CIMS into an NO+ ToF-CIMS, and a solid discussion of instrument sensitivities and challenges in measuring a suite of volatile organic compounds in the atmosphere. The Supplemental contains relevant and useful information. This manuscript is appropriate for publication in this journal. While the paper is comprehensive and well-written, I have two major comments that the authors should address.

Major comments. 1. The authors describe differences in sensitivity between NO+ and H3O+ ionization. However, it is unclear how this parameter of sensitivity was determined. Typically sensitivity is taken as the slope of a calibration curve (i.e. different signals of a VOC as measured by an instrument as a function of different concentrations). From the manuscript, I think that the authors instead only used a single concentration (they write "each VOC was sampled twice, once with H3O+ and once with NO+"). This is unfortunately not representative of instrument sensitivity, as this approach assumes linearity - which is often not the case for CIMS measurements. Did the authors do a proper multi-point calibration, or just take a single point? In either case, to what extent is the instrument linear for the selected analytes across an atmospherically relevant range? Finally, to what extent is the instrument response linear for the selected analytes across an atmospherically relevant range of relative humidity? As the charging mechanisms clearly change as a function of RH, I would not be surprised if the changes in charging mechanisms shifted at higher RH leading to non-linearities in sensitivity. In my opinion, linearity (or a clear understanding of non-linearity!) is an essential parameter to demonstrate when proving a new instrument is valuable for atmospheric measurements.

2. The discussion of NO+ mechanisms would benefit from more detail. In lines 220-224, the authors describe charging mechanisms for the NO+ reagent ion. Are these statements based on previous work (in which case, references are required), or this work (in which case, more evidence is required). The authors present two particularly useful pieces of information: Table 2, in which dominant peaks observed by NO+ CIMS are described along with sensitivities, and Figure 4, in which the theoretical basis of the charging mechanisms are quantified. While the observed signals for most components (e.g. methanol and benzene) appear to clearly fit the theory, it looks like some molecules may not. If I'm reading the figure and tables correctly, toluene (molecule 41) looks like it should be charged approximately equally by a charge transfer and hydride reaction. However, Table 2 suggests that toluene is charged almost entirely via a charge transfer from the NO+. As the instrument should be in an equilibrium-dominated regime for ion-molecule interactions, as opposed to a kinetically-limited regime, this is surprising. To what extent are the observed ions consistent with the predicted distribution? Can differences be attributed to changes in transmission efficiency as a function of m/z, or the breaking of adducts downstream in the mass spectrometer?

Minor comments.

line 155-156. Replace "10e6" with proper scientific notation

Figure 2E: Figure caption should note the RH at which experiments were performed, as the signals for the NOH2O+ cluster, H3O+ and H2OH3O+ cluster should depend on that.

---

## Author Comment (AC1) · 18 Jun 2016

Journal: AMT
Title: Evaluation of NO$^+$ reagent ion chemistry for on-line measurements of atmospheric volatile organic compounds
Authors: A. R. Koss et al.
MS No.: amt-2016-78
MS Type: Research Article

**Response to reviewer 3**
**Reviewer comments are in black text.**
Our response is in blue text.
**Changes to the manuscript are in bold text.**

This manuscript presents a fairly straight-forward way to adapt a PTR-TOF mass spectrometer to use NO+ chemical ionization. It presents a comprehensive ensemble of experiments to establish the involved ion chemistry, and the methods employed. It also offers a thorough discussion of the results and the performance obtained, in particular in comparison to widely used PTR instrumentation. Furthermore, the results are very clearly presented (although I suspect there is still a bit of room for improvement). Therefore, I think that the manuscript is very well within the scope of AMT, and I believe it will be of high interest to the chemical ionization mass spec community. I recommend its publication, with a few minor revisions or clarifications.

We thank the reviewer for their detailed and helpful comments. Replies to specific comments are below.

Specific comments:
Section 2 (e.g. lines 139+): How were the quadrupole ion guides set? As the authors state, these ion guides can change the measured ion distributions, e.g. via changing the m/z-dependent transmission, and adduct products can be more or less strongly de-clustered, directly affecting the presented results. (If one so desired, spectra could indeed be altered quite drastically.) So I wonder, was there any tuning done on the ion guidance elements, specifically for the presented experiments? Is it possible to compare the used quad settings with other conditions tested by the authors, or by others?

The reviewer raises an important point that the ion guide elements can substantially change measured distributions of VOC ions. In particular, VOC·NO+ adducts can be declustered by non-optimal ion guide settings.

At the end of this section (line 139+, line 171 in edited manuscript), we have added more explanation:
**"The two most important such adjustments decreased the electric potentials immediately upstream of each quadrupole ion guide (Figure S2). These adjustments reduced declustering at these locations, which improved the transmission of VOC·NO+ clusters."**

**For transparency, and for readers interested in the details of the settings, we have added Figure S2 to the supplementary information.** This figure compares the ion guide voltage settings between the H$_3$O$^+$ and NO$^+$ ToFCIMS. The optimal values of these voltages are instrument dependent, but the figure highlights the most important changes.

Table 2: Not totally clear to me, what the gray text means. The table caption says that gray is for additional product ions. "Additional" apparently meaning that they were not used to establish an NO+ sensitivity. I assume the last column texts are gray as well, because those detection limits do not directly relate to any

of the numbers or formulas to the left. Or is there any relation to the other gray text? Could that be somehow presented in a clearer manner?

**We changed the color of the "H3O+ CIMS detection limit" column to black and edited Table 2 caption to read, "Additional product ions not used to establish sensitivity are listed in gray text. The H$_3$O$^+$ ToF-CIMS detection limits in the farthest right column are calculated from separate H$_3$O$^+$ ToF-CIMS calibrations as decribed in Yuan et al. (2016)". We also added a vertical line to visually separate the H$_3$O$^+$ CIMS values from the rest of the table, which is exclusively NO$^+$ chemistry.**

Sections 2 and 3.1.1-3.1.4 suggest that sensitivities (and simplicities) as well as reaction and fragmentation patterns were determined at dry conditions (I think that could be stated more clearly), whereas section 3.1.5 discusses humidity dependence. There seem to be some inconsistencies comparing some Figures and Tables. For instance taking MVK: Table 2B lists the product ions to be 100% C4H6ONO+ (m/z 100), but Fig. 2A, at operating voltages, shows that the product ion at m/z 100 (presumably C4H6ONO+) constitutes only ~70% of product ions. Other example, isoprene: Table 2A lists 46% of the signal as C5H8+, 17% as C5H8NO+, 7% as C5H7+, whereas Fig. 7B, at dry conditions, suggest >90% of the signal is C5H8+, <10% from C5H8NO+ and C5H9+ (not listed in Table 2, while C5H7+ is not shown in Fig. 7B). There is a similar discrepancy for benzene, where Table 2A doesn't agree with Fig. 7D, except maybe at about 20% RH... I may have gotten confused with the differences in experimental setup/conditions, or I understand Table 2 wrong? Either way, some clarification would be helpful, at least for this reviewer. Maybe by an additional table that summarizes conditions for each figure's/table's experiments?

This is a very helpful comment, because the discrepancies between product ion distributions reported in different sections of the paper are indeed due to different experimental setup/conditions, and some clarification is absolutely needed.

To clarify: The relevant difference in experimental condition was the relative humidity. In the laboratory, two relative humidities were used: dry air, and 20%.
The experiments done in dry air were: Adaptation of H$_3$O$^+$ to NO$^+$ CIMS (Section 2.2, Figure 1, Figure 2).
The 20% relative humidity is in the typical range of ambient relative humidity at our sampling site and time of year. By choosing a laboratory relative humidity condition similar to that expected for ambient conditions, it is easier and more robust to use the laboratory results to interpret the ambient data. The experiments done in 20% relative humidity or ambient humidity were: Laboratory experiments (Section 3.1, Table 2, Figures 3 through 6); and Measurements of urban air (Section 3.2, Table 2, Table 3, Figures 8 through 12).

How we have addressed this in the manuscript:
**In the description of each experiment, and in each table and figure, we have ensured there is a short statement of the humidity condition used.**
> **At line 181: "A relative humidity of 20% was used for this experiment. This humidity condition is similar to that expected for ambient measurements discussed in Section 3.2; this condition was chosen to aid interpretation of ambient air data. Humidity effects are discussed in Section 3.1.5."**
> **At line 406: "The NO$^+$ ToF-CIMS was calibrated using air with ambient humidity (approximately 20%) for the 10 species listed in Table 2A, and no further humidity correction was applied."**
> **Table 2: ambient RH indicated for each section of the table.**
> **Figure 1: Added "Experiment conducted in dry air (H$_3$O$^+$ is from residual water in the instrument and in commercial ultrazero air.)"**

**Figure 2:** Added "Experiment conducted in dry air ($H_3O^+$ is from residual water in the instrument and in commercial ultrazero air.)"
**Figure 3:** relative humidity 20% specified.
**Figure 5:** relative humidity 20% specified.
**Figure 6:** relative humidity 20% specified.
**Figure 7B:** m/z 67 $C_5H_7$ added.
**Figures 10, 11, 12:** ambient air specified.
**Figure S4:** relative humidity 20% specified.

**Finally, we have updated Table 2 to be entirely internally consistent: all $NO^+$ sensitivities and product ion distributions reported here (in the revised manuscript) are now determined at ambient (20%) relative humidity.** Originally, the sensitivities of the 11 compounds reported in Table 2A were determined in dry air, while all product ion distributions, and the sensitivities in sections 2B and 2C were determined in humid air. This was confusing and not indicated clearly. The new sensitivities in section 2A are calculated using data from multiple-step calibrations and background measurements conducted in air of ambient humidity (~20%). The revised limits of detection are not significantly different. **The revision to how limits of detection were calculated has been indicated at line 293.**

Line 332: How long was a "measurement period"? And Section 3.2 in general: How did the zero measurements look like? Was the frequency of zero measurements sufficient? The frequency was once every hour for 3.2.2, but could not find out for 3.2.1 (see comment on line 332). I'd assume it was enough, but maybe show in supplement, by a figure, or by giving a few numbers.

"Measurement period" is the full three-day experiment. **To clarify, we have added the qualifier "three-day" to "measurement period" in all places where it appears in the manuscript. We also clarify that the 56-component standard was also used to determine instrument background on a daily basis (line 372).**

The background of the $NO^+$ ToF-CIMS is of more interest to the CIMS community than the GC-interface CIMS. **To show the background of the NO+ ToF-CIMS, we have added Figure S10, which shows a time series of count rate, including background measurements, for several compounds.** The ion masses selected are m/z 88 $C_3H_6ONO$, m/z 71 $C_5H_{11}$, and m/z 78 $C_6H_6$ to show background for a range of functional groups, $NO^+$ ionization mechanisms, and signal intensity. The zeros are clearly of sufficient frequency to establish background stability.

Fig. S8 gives a glimpse at the challenges of TOF data analysis, in this case overlapping peaks, requiring "high-resolution peak-fitting algorithms" (line 422). I was missing some short statement on what software was used to tackle these and other challenges.

**The name and manufacturer of the peak-fitting software (Tofware, from Aerodyne/Tofwerk AG), and a citation of a description of the algorithm (DeCarlo et al., 2006), have been added to the methods section in 2.1.**

Line 141: "does not have that issue as strongly" is a bit too vague. I would prefer a more quantitative statement, or a suitable reference.

Modeled (expected) reagent ion distributions in the PTR-QMS -- based on ion energetics and relative humidity -- have been compared to actual, measured distributions, and the two distributions are quite similar (de Gouw and Warneke, 2007). Therefore, there is experimental evidence that the ion guides and

mass analyzer in the PTRQMS do not significantly change the reagent ion and cluster distribution. The difference between the two instruments is not surprising because the PTR-QMS does not have quadrupole ion guides, while the ToF CIMS does. **We have added a reference to de Gouw and Warneke (2007) with a short explanation at line 141 (line 143 in edited manuscript).**

Line 354: I just don't get the message of this sentence into my head. I suggest the authors break it down or reformulate.

What we meant by this statement is that we can identify ion masses that have contributions from multiple VOCs by comparing the $NO^+$ ToF-CIMS and GC-ToF-CIMS measurements. If an interference comes from a VOC that cannot be transmitted through the GC, then the $NO^+$ ToF-CIMS will measure a higher signal (the additional signal comes from the interfering VOC), and higher variability (if the GC-transmittable VOC and the interfering VOC are not perfectly correlated). **On reflection, this statement does not contain any useful ideas that are not already in this paragraph, so we have deleted line 354 (line 393 in edited manuscript).**

Technical corrections:
Line 11: "fast (sub 1-Hz)" ... To my understanding "sub 1-Hz" suggests slower than 1 Hz. Is "sub 1-Hz" what is really meant here, i.e. "up to 1 Hz"? Or the opposite, i.e. "1 Hz and faster"? The paper does present experiments at 1Hz and slower, but I would think faster (>1Hz) measurements would be possible as well.

The instrument is capable of measurements faster than 1 Hz, although we only present 1 Hz measurement here. **To clarify, we replaced "sub 1 Hz" with "1 Hz and faster" as suggested, and made similar edits elsewhere in the manuscript where "sub-" refers to measurement frequency.**

**References**
de Gouw, J., and Warneke, C.: Measurements of volatile organic compounds in the earth's atmosphere using proton-transfer-reaction mass spectrometry, Mass. Spectrom. Rev., 26, 223-257, 10.1002/mas.20119, 2007.

DeCarlo, P. F., Kimmel, J. R., Trimborn, A., Northway, M. J., Jayne, J. T., Aiken, A. C., Gonin, M., Fuhrer, K., Horvath, T., Docherty, K. S., Worsnop, D. R., and Jimenez, J. L.: Field-Deployable, High-Resolution, Time-of-Flight Aerosol Mass Spectrometer, 78, 8281-8289, 10.1021/ac061249n, 2006.

[Figure]

**Updated Figure 7**

[Figure]

**Figure S2. Ion guide voltage settings. The top panel shows the absolute voltage setting (from ground); the middle panel highlights the changes in voltage potential between H₃O⁺ and NO⁺ settings, and the bottom panel is a cartoon of the ion guide section taken from the CI-API manual (Aerodyne Inc./Tofwerk AG). The horizontal (axial) distances are not to scale.**

[Figure]

**Figure S10. A. Background and ambient measurements taken during urban air sampling with the NO⁺ ToF-CIMS. B. Example multiple-point calibrations of the NO⁺ ToF-CIMS showing sensitivity linear with concentration.**

---

## Author Comment (AC2) · 18 Jun 2016

Journal: AMT
Title: Evaluation of NO⁺ reagent ion chemistry for on-line measurements of atmospheric volatile organic compounds
Authors: A. R. Koss et al.
MS No.: amt-2016-78
MS Type: Research Article

**Response to Short Comment by S. Inomata**
The short comment is in black text.
Our response is in blue text.

Our group has reported product ions and their relative intensities of C3-C13 n-alkanes and C4-C10 iso-alkanes measured by NO+ CIMS (see Table A1 of the Supplementary Data in Yamada et al. (2015)). The present results are generally similar to ours. But it seems that the ratios of fragment ions to [M-H]+ ions are larger in the present study than ours. I think that the strength of the electric field of the drift tube (i.e., E/N ratio) cannot be the reason of this difference because the ratios are similar in both the studies (60 Td in the present study and 67 Td in our study). We also showed that O2+ ionization of alkanes produces the same fragment ions as NO+ ionization (see Table A1 of the Supplementary Data in Yamada et al. (2015)). Therefore, we subtracted the contribution of O2+ ionization from ion signals in order to report the detection sensitivities of alkanes by NO+ CIMS. Did the authors consider the contribution of O2+ ionization when they measured alkanes by NO+ CIMS?

Yamada, H., Inomata, S., and Tanimoto H.: Evaporative emissions in three-day diurnal breathing loss tests on passenger cars for the Japanese market, Atmos. Environ., 107, 166-173, 2015.

We are pleased to see a short comment from Dr. Inomata – his work (Inomata et al., 2013) has motivated us to investigate high-mass alkanes.

On comparing the contaminant $O_2^+$ reported in our work (4% of NO⁺ signal) to that reported in Yamada et al. (2015) (1.5% of NO⁺ signal), it seems likely that the difference in $O_2^+$ contributes to the difference in the reported product ion distributions. We have acknowledged this important consideration in two places in the revised manuscript. First, in section 3.1.2 ("Distribution of Product Ions"), we have included a comparison to Yamada et al. tridecane distribution in Figure 5, along with an explanation (see our response to Reviewer 1 for details). Second, in section 3.1.3 ("Alkane Fragmentation"), at line 297, we have added a brief consideration of the effect of contaminant $O_2^+$ on sensitivity and product ion distribution.

We do not have a characterization of our instrument's response to $O_2^+$ ionization, so it is not possible for us to perform a sensitivity correction as in Yamada et al.

**References**
Blake, R. S., Wyche, K. P., Ellis, A. M., and Monks, P. S.: Chemical ionization reaction time-of-flight mass spectrometry: Multi-reagent analysis for determination of trace gas composition, Int. J. Mass Spectrom., 254, 85-93, http://dx.doi.org/10.1016/j.ijms.2006.05.021, 2006.
Inomata, S., Tanimoto, H., and Yamada, H.: Mass Spectrometric Detection of Alkanes Using NO+ Chemical Ionization in Proton-transfer-reaction Plus Switchable Reagent Ion Mass Spectrometry, Chem. Lett., 43, 538-540, 10.1246/cl.131105, 2013.

Španěl, P., Ji, Y., and Smith, D.: SIFT studies of the reactions of H3O+, NO+ and O2+ with a series of aldehydes and ketones, Int. J. Mass Spectrom., 165–166, 25-37, http://dx.doi.org/10.1016/S0168-1176(97)00166-3, 1997.

Španěl, P., and Smith, D.: Selected ion flow tube studies of the reactions of H3O+, NO+, and O2+ with several aromatic and aliphatic hydrocarbons, Int. J. Mass Spectrom., 181, 1-10, http://dx.doi.org/10.1016/S1387-3806(98)14114-3, 1998.

Wyche, K. P., Blake, R. S., Willis, K. A., Monks, P. S., and Ellis, A. M.: Differentiation of isobaric compounds using chemical ionization reaction mass spectrometry, Rapid Commun. Mass Spectrom., 19, 3356-3362, 10.1002/rcm.2202, 2005.

Yamada, H., Inomata, S., and Tanimoto, H.: Evaporative emissions in three-day diurnal breathing loss tests on passenger cars for the Japanese market, Atmos. Environ., 107, 166-173, http://dx.doi.org/10.1016/j.atmosenv.2015.02.032, 2015.

[Figure]

**Figure 5. Comparison of product ion distributions between four sets of instrumental and environmental conditions.**
*a. Španěl and Smith (1998a)*
*b. Blake et al. (2006)*
*c. Španěl et al. (1997)*
*d. Wyche et al. (2005)*
*e. Yamada et al. (2015)*

---

## Author Comment (AC3) · 18 Jun 2016

**Journal: AMT**
**Title: Evaluation of NO+ reagent ion chemistry for on-line measurements of atmospheric volatile organic compounds**
**Authors: A. R. Koss et al.**
**MS No.: amt-2016-78**
**MS Type: Research Article**

**Response to reviewer 1**
Reviewer comments are in black text.
Our response is in blue text.
**Changes to the manuscript are highlighted in bold text.**

This is a well-written and detailed description of the conversion of an H3O+ ToF-CIMS into an NO+ ToF-CIMS, and a solid discussion of instrument sensitivities and challenges in measuring a suite of volatile organic compounds in the atmosphere. The Supplemental contains relevant and useful information. This manuscript is appropriate for publication in this journal. While the paper is comprehensive and well-written, I have two major comments that the authors should address.

We thank the reviewer for their positive and insightful comments. Responses to specific questions are below.

Major comments.
1. The authors describe differences in sensitivity between NO+ and H3O+ ionization. However, it is unclear how this parameter of sensitivity was determined. Typically sensitivity is taken as the slope of a calibration curve (i.e. different signals of a VOC as measured by an instrument as a function of different concentrations). From the manuscript, I think that the authors instead only used a single concentration (they write "each VOC was sampled twice, once with H3O+ and once with NO+"). This is unfortunately not representative of instrument sensitivity, as this approach assumes linearity - which is often not the case for CIMS measurements. Did the authors do a proper multi-point calibration, or just take a single point? In either case, to what extent is the instrument linear for the selected analytes across an atmospherically relevant range? Finally, to what extent is the instrument response linear for the selected analytes across an atmospherically relevant range of relative humidity? As the charging mechanisms clearly change as a function of RH, I would not be surprised if the changes in charging mechanisms shifted at higher RH leading to non-linearities in sensitivity. In my opinion, linearity (or a clear understanding of non-linearity!) is an essential parameter to demonstrate when proving a new instrument is valuable for atmospheric measurements.

First, we will respond to the reviewer's concerns about non-linearity of sensitivity with concentration. We can assume linear sensitivity over the investigated concentration range in this situation for the following reasons:
▪ First, we did multi-point calibrations across an atmospherically relevant concentration range to generate the quantitative sensitivities for the 11 compounds reported in Table 2A. All 11 species, which include a range of functional groups, had linear sensitivity through the entire range (0-10 ppb). We added an extra figure to the supplement (Fig. S10) to show this. (The addition of Fig. S10 is noted in the "changes to manuscript", below.)
▪ Second, $H_3O^+$ CIMS (PTRMS) instruments consistently have linear sensitivity over a very wide range of concentrations (de Gouw and Warneke, 2007;Sulzer et al., 2014) and we have demonstrated

linearity in our $H_3O^+$ CIMS system in a separate publication (Yuan et al., 2016). We are therefore not concerned about non-linearity in the $H_3O^+$ data used to generate Figure 3.

▪ Third, we compared $NO^+$ ToF-CIMS measurements of ambient air to independent GC-MS measurement, and there is good agreement ($R^2$ typically >0.9, Table 3, Section 3.2.2).

The data used to generate Figure 3 (comparison of $NO^+$ and $H_3O^+$ sensitivity) were collected using a single concentration data point each for $NO^+$ and for $H_3O^+$. There is an example calculation shown in Figure S5. The calculation involves integrating the area under a GC peak, which inherently accounts for the instrument background. Essentially, we assumed a linear sensitivity extrapolated from two points: the instrument response under the GC peak, and the instrument response off the GC peak.

The ideal way to create Figure 3 would have been to quantitatively introduce each of the 87 species shown in Figure 3 into the $NO^+$ ToF-CIMS, do a multiple-point calibration, then repeat each multiple-point calibration with $H_3O^+$ settings for a total of 174 individual calibrations. This is just too time consuming for the point we want to make in Figure 3. We note that finding methods to quickly and inexpensively calibrate hundreds of species detected by CIMS instruments is a broad current challenge in the CIMS community.

Next, we will respond to the reviewer's concerns about non-linearity of sensitivity with changing humidity. This is a very valid concern. Determining a thorough, robust, and easily applicable treatment of humidity effects in $NO^+$ CIMS will require much careful work and we anticipate that it will be the subject of a separate manuscript, just as humidity effects in PTRMS were discussed in earlier focused investigations (e.g. Španěl and Smith (2000)). We would like to highlight that there is good agreement between the $NO^+$ ToF-CIMS and GCMS measurements of ambient air (Table 3). The good agreement between the two independent techniques, despite the use of $NO^+$ ToF CIMS data that is not humidity corrected, is a very promising result when evaluating the $NO^+$ technique for atmospheric measurements. This indicates that the humidity effect is likely not severe for most species.

We have incorporated this discussion into the manuscript in the following ways:

**We have added Figure S10 to the supplemental information. This figure shows the periodic zeros taken during ambient air measurements (see response to Reviewer 3) and a linear sensitivity from multiple-point calibrations for several VOCs.**

At line 194 in the edited manuscript, we have added,

**"Because only one concentration was sampled, this metric relies on sensitivity being linear with concentration. Linear sensitivity is assumed for the $NO^+$ and $H_3O^+$ ToF-CIMS because separate multiple-point calibrations for select VOCs showed a linear response (Section 3.1.4), $H_3O^+$ CIMS has demonstrated linear sensitivity over a wide range of concentrations (de Gouw and Warneke, 2007;Sulzer et al., 2014;Yuan et al., 2016), and the $NO^+$ CIMS agrees well with an independent technique over a range of atmospheric concentrations (Section 3.2.2)."**

At line 415 in the edited manuscript, we have added.

**"The good agreement also indicates that humidity-dependence of sensitivity is likely not a severe effect for most species; however, addressing and quantifying this effect should be a priority for future work."**

2. The discussion of NO+ mechanisms would benefit from more detail. In lines 220-224, the authors describe charging mechanisms for the NO+ reagent ion. Are these statements based on previous work (in which case, references are required), or this work (in which case, more evidence is required). The authors present two particularly useful pieces of information: Table 2, in which dominant peaks observed by NO+

CIMS are described along with sensitivities, and Figure 4, in which the theoretical basis of the charging mechanisms are quantified. While the observed signals for most components (e.g. methanol and benzene) appear to clearly fit the theory, it looks like some molecules may not. If I'm reading the figure and tables correctly, toluene (molecule 41) looks like it should be charged approximately equally by a charge transfer and hydride reaction. However, Table 2 suggests that toluene is charged almost entirely via a charge transfer from the NO+. As the instrument should be in an equilibrium-dominated regime for ion-molecule interactions, as opposed to a kinetically-limited regime, this is surprising. To what extent are the observed ions consistent with the predicted distribution? Can differences be attributed to changes in transmission efficiency as a function of m/z, or the breaking of adducts downstream in the mass spectrometer?

We included Figure 4 to provide a framework to explain why certain groups of VOCs undergo particular ionization mechanisms, and to suggest a likely mechanism for VOCs not explicitly studied in this work. We do not want to suggest that Figure 4 is an absolute predictor of ionization mechanism. The reaction enthalpies of charge transfer and hydride abstraction need to be such that the reactions are thermodynamically allowed, but they do not predict the relative rate coefficients for the two processes. Similar behavior is observed in PTRMS ($H_3O^+$ CIMS), where the product ions are not in equilibrium with the primary ions, and the number of product ions are controlled by the rate constants of the proton-transfer reactions and the average time the ions spend in the drift tube (Lindinger et al., 1998). A more complete understanding of all ionization mechanisms is well beyond the scope of this work.

The statements in this paragraph (220-224, lines 228-242 in edited manuscript) are a description of the patterns shown in Figure 4. The thermodynamic information is available in reference tables from Lias et al. (1988). The mechanistic information is taken from this work and from a large collection of SIFT studies. Our understanding of $NO^+$ ionization mechanisms relies heavily on excellent and extensive work done by especially Smith, Španěl, and colleagues using SIFT techniques and we provide more citations in the revised manuscript.

The product ion distribution depends also on instrumental conditions. There are differences in measured product ion distributions between our work and previous work and these are likely due to a combination of E/N settings in the ion-molecule reaction region and the presence of impurity ions such as $O_2^+$. Mass-dependent transmission is an unlikely explanation, as transmission is nearly mass-independent above m/z 30 (Yuan et al., 2016).

Regarding the reviewer's questions about toluene (as an example), we realize that the 1:1 line in Figure 4 seems to suggest that molecules on this line should equally participate in charge transfer and hydride abstraction reactions. However, the actual ionization mechanism depends on more than simple thermodynamics, as stated above. To avoid further confusion **we have removed the 1:1 line from Figure 4.**

We have revised section 3.1.2 to address the limitations of Figure 4 and added several relevant citations. The revised Section 3.1.2 now reads:
        **"It is somewhat more difficult to predict the ionized VOC products of $NO^+$ CIMS compared to $H_3O^+$ CIMS, because $NO^+$ has three common reaction mechanisms: charge transfer, hydride abstraction, and cluster formation. Groups of VOCs that have similar charge transfer and hydride abstraction enthalpies tend to react with similar ionization mechanisms (Figure 4). Figure 4 uses thermodynamic information from Lias et al. (1988), and mechanistic information from this work (see table S1 for a list of species) and from SIFT studies (Španěl and Smith, 1996, 1998a, b, 1999;Španěl et al., 1997;Arnold et al., 1998;Francis et al., 2007a;Francis et al., 2007b). Charge transfer occurs if the reaction enthalpy is**

favorable, regardless of the hydride transfer enthalpy. If the charge transfer enthalpy is close to zero, then NO$^+$ clustering occurs; and if charge transfer is not favorable but hydride transfer is, then hydride transfer will occur. In terms of VOC families, this means that carbonyls participate in two mechanisms: ketones cluster with NO$^+$, and aldehydes hydride transfer. Branched alkanes exclusively undergo hydride transfer. Aromatics undergo charge transfer and benzene also clusters; alcohols undergo hydride transfer, and alkenes charge transfer, cluster, or hydride transfer depending on the size of the molecule and the location of the double bond within the molecule.

Although Figure 4 provides a general way to predict the possible mechanisms for a particular VOC, it provides no information about the distribution of the signal between different mechanisms or the degree of fragmentation. The distribution depends strongly on instrumental conditions, which include E/N settings in the ion-molecule reaction region, which is by far the most important effect, fragmentation and clustering in the ion optics, presence of impurity ions such as O$_2^+$ from the converted hollow cathode ion source, and relative humidity (Section 3.1.5).

In Figure 5 the product ion distributions of several VOCs determined in this work are compared to three others using NO$^+$. Studies by the University of Leicester used a much higher E/N ratio in the drift tube, leading to higher fragmentation and lower NO$^+$ adduct formation compared to this work (Wyche et al., 2005;Blake et al., 2006). Investigation of higher-mass alkanes by Yamada et al. (2015)used similar E/N, but achieved lower contaminant O$_2^+$, which is a likely explanation for the higher degree of fragmentation of tridecane seen in this work. In SIFT-MS studies, without an electric field, fragmentation is minimized and preselection of NO$^+$ primary ions eliminates contaminant H$_3$O$^+$ and O$_2^+$ and therefore SIFT product ion distributions are generally simpler. These differences highlight the importance of selection of drift tube operating conditions and instrument characterization."

We have edited the caption of Figure 4 to read "Ion thermodynamic information is available for several species whose reaction mechanism was not experimentally verified in this or previous work …"

We have added Figure 5, which compares product ion distributions from several laboratory studies of NO$^+$ CIMS. To reduce the number of figures in the manuscript, we have moved the original Figure 5 (example aliphatic product ion distributions) to the supplementary material (now Figure S4).

Minor comments.
line 155-156. Replace "10e6" with proper scientific notation
        Fixed
Figure 2E: Figure caption should note the RH at which experiments were performed, as the signals for the NOH2O+ cluster, H3O+ and H2OH3O+ cluster should depend on that.
        This is addressed in our response to the other reviewer.

**Citations**
Arnold, S. T., Viggiano, A. A., and Morris, R. A.: Rate Constants and Product Branching Fractions for the Reactions of H3O+ and NO+ with C2–C12 Alkanes, J. Phys. Chem. A, 102, 8881-8887, 10.1021/jp9815457, 1998.
Blake, R. S., Wyche, K. P., Ellis, A. M., and Monks, P. S.: Chemical ionization reaction time-of-flight mass spectrometry: Multi-reagent analysis for determination of trace gas composition, Int. J. Mass Spectrom., 254, 85-93, http://dx.doi.org/10.1016/j.ijms.2006.05.021, 2006.
de Gouw, J., and Warneke, C.: Measurements of volatile organic compounds in the earth's atmosphere using proton-transfer-reaction mass spectrometry, Mass. Spectrom. Rev., 26, 223-257, 10.1002/mas.20119, 2007.

Francis, G. J., Milligan, D. B., and McEwan, M. J.: Gas-Phase Reactions and Rearrangements of Alkyl Esters with H3O+, NO+, and O2•+: A Selected Ion Flow Tube Study, J. Phys. Chem. A, 111, 9670-9679, 10.1021/jp0731304, 2007a.

Francis, G. J., Wilson, P. F., Milligan, D. B., Langford, V. S., and McEwan, M. J.: GeoVOC: A SIFT-MS method for the analysis of small linear hydrocarbons of relevance to oil exploration, Int. J. Mass Spectrom., 268, 38-46, 10.1016/j.ijms.2007.08.005, 2007b.

Lias, S. G., Bartmess, J. E., Liebman, J. F., Holmes, J. L., Levin, R. D., and Mallard, W. G.: Gas-Phase Ion and Neutral Thermochemistry, J. Phys. Chem. Ref. Data, 17, 1988.

Lindinger, W., Hansel, A., and Jordan, A.: On-line monitoring of volatile organic compounds at pptv levels by means of proton-transfer-reaction mass spectrometry (PTR-MS) - Medical applications, food control and environmental research, Int. J. Mass Spectrom., 173, 191-241, 10.1016/s0168-1176(97)00281-4, 1998.

Španěl, P., and Smith, D.: A selected ion flow tube study of the reactions of NO+ and O+2 ions with some organic molecules: The potential for trace gas analysis of air, J. Chem. Phys., 104, 1893-1899, doi:http://dx.doi.org/10.1063/1.470945, 1996.

Španěl, P., Ji, Y., and Smith, D.: SIFT studies of the reactions of H3O+, NO+ and O2+ with a series of aldehydes and ketones, Int. J. Mass Spectrom., 165–166, 25-37, http://dx.doi.org/10.1016/S0168-1176(97)00166-3, 1997.

Španěl, P., and Smith, D.: Selected ion flow tube studies of the reactions of H3O+, NO+, and O2+ with several aromatic and aliphatic hydrocarbons, Int. J. Mass Spectrom., 181, 1-10, http://dx.doi.org/10.1016/S1387-3806(98)14114-3, 1998a.

Španěl, P., and Smith, D.: Selected ion flow tube studies of the reactions of H3O+, NO+, and O2+ with several amines and some other nitrogen-containing molecules, Int. J. Mass Spectrom., 176, 203-211, http://dx.doi.org/10.1016/S1387-3806(98)14031-9, 1998b.

Španěl, P., and Smith, D.: Selected ion flow tube studies of the reactions of H3O+, NO+, and O2+ with several aromatic and aliphatic monosubstituted halocarbons, Int. J. Mass Spectrom., 189, 213-223, http://dx.doi.org/10.1016/S1387-3806(99)00103-7, 1999.

Španěl, P., and Smith, D.: Influence of water vapour on selected ion flow tube mass spectrometric analyses of trace gases in humid air and breath, 14, 1898-1906, 10.1002/1097-0231(20001030)14:20<1898::AID-RCM110>3.0.CO;2-G, 2000.

Sulzer, P., Hartungen, E., Hanel, G., Feil, S., Winkler, K., Mutschlechner, P., Haidacher, S., Schottkowsky, R., Gunsch, D., Seehauser, H., Striednig, M., Jürschik, S., Breiev, K., Lanza, M., Herbig, J., Märk, L., Märk, T. D., and Jordan, A.: A Proton Transfer Reaction-Quadrupole interface Time-Of-Flight Mass Spectrometer (PTR-QiTOF): High speed due to extreme sensitivity, Int. J. Mass Spectrom., 368, 1-5, http://dx.doi.org/10.1016/j.ijms.2014.05.004, 2014.

Wyche, K. P., Blake, R. S., Willis, K. A., Monks, P. S., and Ellis, A. M.: Differentiation of isobaric compounds using chemical ionization reaction mass spectrometry, Rapid Commun. Mass Spectrom., 19, 3356-3362, 10.1002/rcm.2202, 2005.

Yamada, H., Inomata, S., and Tanimoto, H.: Evaporative emissions in three-day diurnal breathing loss tests on passenger cars for the Japanese market, Atmos. Environ., 107, 166-173, http://dx.doi.org/10.1016/j.atmosenv.2015.02.032, 2015.

Yuan, B., Koss, A., Warneke, C., Gilman, J. B., Lerner, B. M., Stark, H., and de Gouw, J. A.: A high-resolution time-of-flight chemical ionization mass spectrometer utilizing hydronium ions (H3O+ ToF-CIMS) for measurements of volatile organic compounds in the atmosphere, Atmos. Meas. Tech. Discuss., 2016, 1-43, 10.5194/amt-2016-21, 2016.

[Figure]

**Figure S10. A. Background and ambient measurements taken during urban air sampling with the NO⁺ ToF-CIMS. B. Example multiple-point calibrations of the NO⁺ ToF-CIMS showing sensitivity linear with concentration.**

[Figure]

**Figure 5. Comparison of product ion distributions between four sets of instrumental and environmental conditions.**
*a. Španěl and Smith (1998a)*
*b. Blake et al. (2006)*
*c. Španěl et al. (1997)*
*d. Wyche et al. (2005)*
*e. Yamada et al. (2015)*

[Figure]

**Revised Figure 4**